

# Vertical profiles and surface distributions of trace gases (CO, O₃, NO, NO₂) in the Arctic wintertime boundary layer using low-cost sensors during ALPACA-2022

Brice Barret [1], Patrice Medina [1], Natalie Brett [2], Roman Pohorsky [3], Kathy Law [2], Slimane Bekki [2], Gilberto J. Fochesatto [4], Julia Schmale [3], Steve Arnold [5], Andrea Baccarini [3], Maurizio Busetto [6], Meeta Cesler-Maloney [7], Barbara D'Anna [8], Stefano Decesari [6], Jingqiu Mao [7], Gianluca Pappaccogli [9], Joel Savarino [10], Federico Scoto [9], and William Simpson [7]

[1] Laboratoire d'Aérologie, Université Paul Sabatier, Université de Toulouse - CNRS, France
[5] School of Earth and Environment, University of Leeds, Leeds, LS2 9JT, UK
[3] Extreme Environments Research Laboratory, Ecole Polytechnique Fédérale de Lausanne (EPFL), 1950 Sion, Switzerland
[2] Laboratoire Atmosphères, Milieux, Observations Spatiales, Sorbonne Université / Université de Versailles Saint Quentin / CNRS
[6] CNR-ISAC, National Research Council of Italy, Institute of Atmospheric Sciences and Climate, Via Gobetti 101, Bologna, IT 40129, Italy
[7] Geophysical Institute and Department of Chemistry and Biochemistry, University of Alaska Fairbanks, Fairbanks, AK 99775
[8] Aix-Marseille Université - Campus Marseille Centre, BATIMENT DE CHIMIE,3 Place Victor Hugo, Marseille, 13003
[4] Department of Atmospheric Sciences, University of Alaska Fairbanks, Fairbanks, Alaska USA 99775
[9] CNR-ISAC, National Research Council of Italy, Institute of Atmospheric Sciences and Climate, Lecce, Italy
[10] Univ. Grenoble Alpes, CNRS, Institut des Géosciences de l'Environnement (IGE), UMR 5001, Grenoble 38041, France

**Correspondence:** Brice Barret (brice.barret@aero.obs-mip.fr)

**Abstract.**

Electrochemical gas sensors (EGSs) have been used to measure the surface distributions and vertical profiles of trace gases in the wintertime Arctic Boundary Layer during the Alaskan Layered Pollution and Chemical Analysis (ALPACA) field experiment in Fairbanks, Alaska in January-February 2022. The MICRO sensors for MEasurements of GASes (MICROMEGAS)

instrument set up with CO, NO, NO₂ and O₃ EGSs was operated on the ground at an outdoor reference site downtown Fairbanks for calibration, onboard a vehicle moving through the city and its surroundings and onboard a tethered balloon, the Helikite, at a site at the edge of the city. To calibrate the measurements, a set of machine learning (ML) calibration methods were tested. For each method, learning and prediction were performed with coincident MICROMEGAS and reference analyser measurements at the downtown site. For CO, the calibration parameters provided by the manufacturer led to the best agreement

between the EGS and the reference analyser and no ML method was needed for calibration. The correlation coefficient R is 0.82 and the slope of the linear regression between MICROMEGAS and reference data is 1.12. The mean bias is not significant but the Root Mean Square Error (290 ppbv) is rather large because of CO concentrations reaching several ppmv downtown Fairbanks. For NO, NO₂ and O₃, the best agreements for the prediction datasets were obtained with an artificial neural network, the Multi-Layer Perceptron. For these 3 gases, the correlation coefficients are higher than 0.95 and the slopes of linear

regressions with the reference data are in the range 0.93-1.04. The mean biases which are 1±3 ppbv, 0±4 ppbv and 3±12 ppbv



for $NO_2$, $O_3$ and NO respectively are not significant. Measurements from the car round of January 21 are presented to highlight the ability of MICROMEGAS to quantify the surface variability of the target trace gases in Fairbanks and the surrounding hills. MICROMEGAS flew 11 times from the ground up to a maximum of 350 m a.g.l. onboard the Helikite at the site at the edge of the city. The statistics performed over the Helikite MICROMEGAS dataset show that the median vertical gas profiles are

characterised by almost constant mixing ratios. The median values over the vertical are 140, 8, 4 and 32 ppbv for CO, NO, $NO_2$ and $O_3$. Extreme values are detected with low $O_3$ and high $NO_2$ and NO concentrations between 100 and 150 m a.g.l. $O_3$ minimum levels ($5^{th}$ percentile) of 5 ppbv coincident with $NO_2$ maximum levels ($95^{th}$ percentile) of 40 ppbv occur around 200 m a.g.l. The peaks aloft are linked to pollution plumes originating from Fairbanks power plants such as documented with the flight of February 20.

## 25  1  Introduction

Low-cost Electrochemical Gas Sensors (EGSs) have been widely used for air quality (AQ) applications for more than a decade (Karagulian et al. (2019); Kang et al. (2022) and ref therein). Their use is still expanding due to their affordability and the need to fill gaps in existing air quality monitoring networks to better track and understand pollution patterns. However, their calibration is challenging but critical to guarantee their validity and reliability (Kang et al., 2022). Most of the applications take

place with sensors set up on the ground in urban environments at mid-latitudes in the USA (Zimmerman et al., 2018; Casey et al., 2019; Malings et al., 2019), Europe (Mead et al., 2013; Popoola et al., 2016; Spinelle et al., 2015, 2017; Schmitz et al., 2023) or China (Wei et al., 2018; Smith et al., 2019; Liu et al., 2021; Liang et al., 2021). Interestingly, Schmitz et al. (2023) installed low-cost sensors at the ground and at different altitudes on buildings in Berlin streets to document the horizontal and vertical gradients of $O_3$ and $NO_2$ in street canyons. Nevertheless, very few publications deal with the use of EGSs onboard

flying platforms; Li et al. (2017) presented EGS $O_3$ measurements from a Unmanned Aerial Vehicle (UAV) and Schuldt et al. (2023) discussed CO, NO, $NO_2$ and $O_3$ observations from a Zeppelin in Germany. Furthermore, to our best knowledge, low-cost AQ sensors have not been applied in the Arctic region, especially in winter, except for particulate matter and $CO_2$ in the Svalbard archipelago (Carotenuto et al., 2020).

During the winter, extremely low temperatures prevail in the Arctic accompanied by very stable meteorological conditions

and large temperature inversions at the surface (surface-based inversion, SBI) or within the first hundreds of meters above the ground (elevated inversions, EI) (Mayfield and Fochesatto, 2013). Consequently, high emissions from home heating systems and road traffic are trapped near the ground, producing severe air pollution episodes (Schmale et al., 2018). The international Alaskan Layered Pollution and Chemical Analysis (ALPACA) field campaign (Simpson et al., 2024) in January-February 2022, was designed to understand the processes responsible for the regular episodes of poor air quality in Fairbanks. ALPACA

was the first large scale international experiment investigating these issues in the Arctic where anthropisation linked to the exploitation of natural resources (e.g. minerals, energy, marine) and growing human settlements are expected to accelerate with the on-going accelerated warming of the Arctic (Rantanen et al., 2022). As part of ALPACA, outdoor surface observations of trace gases, VOCs and particles were performed at the Community and Technical College (CTC) of the University of Alaska,





Fairbanks (UAF) (64.841°N, 147.727°W; 136 m above sea level), in downtown Fairbanks. The MICRO sensors for MEasure-
ments of GASes (MICROMEGAS) instrument, equiped with NO, NO$_2$, CO and O$_3$ EGSs was deployed at different sites and
on different platforms during the ALPACA campaign. First, MICROMEGAS was regularly operated over periods of hours
to days throughout the campaign at the outdoor CTC site for calibration purposes. It was also used onboard road vehicles to
map surface pollution in and around Fairbanks. As the primary target of the MICROMEGAS deployment in ALPACA, vertical
profiles of trace gases were collected up to 350 m a.g.l with a tethered balloon at the UAF-Farm site at the North-western edge
of Fairbanks (64.853°N, 147.859°W, 138 m above sea level). The tethered balloon combines features of a helium balloon and
a kite to remain stable with winds up to 15 m.s$^{-1}$; it is thereafter referred as Helikite (Pohorsky et al., 2024b).

This novel use of EGSs in extreme cold and polluted conditions onboard moving platforms requires careful calibration and
validation. The EGSs performances are usually found to be very good in the laboratory with controled conditions and gas
concentrations (Mead et al., 2013) but it is challenging to obtain the precisions and accuracies required for AQ applications
in ambient conditions. Indeed, the EGSs ouput voltages show dependences on relative humidity (RH) and temperature (Mead
et al., 2013; Popoola et al., 2016; Spinelle et al., 2015; Liang et al., 2021). Water vapour modifies the equilibrium between
the sampled air and the sensor electrodes and temperature impacts the diffusion of gases into the sensors and the current of
the electrodes (Popoola et al., 2016; Cross et al., 2017; Pang et al., 2018). Cross-sensitivities with trace gases other than the
targeted gas could also be a challenge for EGSs (Kang et al., 2022). Finally, the relationships between the measured voltages
and the impacting atmospheric parameters vary with the range of the ambient conditions (gas concentrations, temperature and
RH). These relationships are subject to changes and drifts when the sensors are used for long periods with changing conditions
or in different locations. Regular measurement periods at the CTC allowed the establishment of a comprehensive database for
MICROMEGAS EGSs calibration throughout the ALPACA-2022 campaign.

The calibration of EGSs is an on-going research area. Results from calibration methods from hundreds of publications about
low-cost sensors have been reviewed by Karagulian et al. (2019) and Kang et al. (2022). To calibrate sensors, one needs to
select the type of calibration data and the calibration method. The calibration can be based on explicit relationships between
the sensor outputs and the ambient parameters derived from laboratory measurements in controled conditions such as in Wei
et al. (2018). Nevertheless, laboratory conditions cannot span all the possible outdoor or indoor conditions and most calibra-
tion methods are derived from field measurements. One also needs to choose the calibration method which provides the best
function to fit reference observations with EGSs measurements. Most of the recent methods used for EGS calibration fall into
the vast field of Machine Learning (ML) (Spinelle et al., 2015, 2017; Bigi et al., 2018; Casey et al., 2019; Malings et al.,
2019; Liang et al., 2021; Bittner et al., 2022). We have therefore tested various ML methods based on the EGS litterature and
selected the best one for each of our target gases. After dealing with the calibration issue, we present an original use of EGSs
to document trace gas distributions in the wintertime Arctic Boundary Layer (ABL).

In section 2 we start by providing details about the observations such as the MICROMEGAS instrument (section 2.1) and the
Modular Multiplatform Compatible Air Measurement System (MoMuCAMS) that hosted this instrument during the balloon
flights (section 2.2). We then describe the reference analysers used for calibration and validation of MICROMEGAS (section
2.3). The operations strategy of MICROMEGAS during the ALPACA-2022 campaign and the different sites where it was



operated are presented in section 2.4. The diverse calibration methods considered here, a crucial element for an adequate use
of EGS, are introduced in section 2.5. Section 3 is dedicated to the presentation of the results starting with the calibration and
validation of the different EGS (CO, NO, $NO_2$, $O_3$) using the reference measurements made at the CTC site (section 3.1). A
comparison with independent measurements performed with instruments from the MoMuCAMS platform at the UAF-Farm
site is presented in section 3.2 for CO and $O_3$. An example of surface mapping of trace gases from on-road mobile sampling
is discussed in section 3.3. The vertical profiles obtained during the tethered balloon flights are finally discussed in section 3.4
and conclusions are presented in section 4.

## 2 Observations

### 2.1 MICROMEGAS instrument

The MICROMEGAS instrument contains A4 (NO, $NO_2$, $O_x=O_3+NO_2$, CO and $SO_2$) EGSs purchased from Alphasense Ltd.
The sensors are set up on 4-sensors AFEs (Analogue Front Ends) which are electronic boards from Alphasense including
amplification and filtering of the sensors signals. The acquisition electronics consists of an ADC (Analogic to Digital Converter)
and a SD card recorder. The EGSs are complemented with 2 Sensirion SHT75 RH and temperature sensors. According to its
datasheet, the SHT75 sensors measure relative humidity and temperature with a 1.8% and 0.3°C accuracy and a 8 s and 5 s
response time respectively. The position of the instrument was recorded from a Diligent Pmod GPS with a 3m 2D satellite
positioning accuracy. Data from the EGSs, temperature and RH sensors and GPS were recorded with a 1 Hz frequency. All
sensors and electronics are set up in a 20x20x15 cm polystryrene box which inside temperature is regulated with a thermal
regulator connected to a thin 4 W film heater. The sensors are set up on specific gas hoods built in PVDF (PolyVinyliDene
Fluoride) provided by Alphasense. The outside air is pumped in with a mini 3.3V diaphragm pump providing a flow of 0.3
l/mn.

### 2.2 MoMuCAMS balloon plateform

The MoMuCAMS is the platform developed to document in situ vertical profiles of aerosol properties, CO, $CO_2$, and $O_3$ in the
lowermost atmosphere flying onboard a tethered balloon, the Helikite (Pohorsky et al., 2024b). MoMuCAMS allows various
combinations of instruments for the observation of multiple aerosol properties (number concentration, size distribution, optical
properties, chemical composition and morphology), as well as CO, $CO_2$, $O_3$ concentrations and meteorological variables (tem-
perature, relative humidity, pressure). CO measurements are performed with a MIRA PICO instrument (Aeris Technologies)
with a precision better than 1 ppbv according to the manufacturer. The $O_3$ instrument is a 2B-Tech monitor with a 1 ppbv (2%)
precision (see Pohorsky et al. (2024b)). The MIRA PICO instrument weighs 2.7 kg and the 2B-Tech $O_3$ monitor weighs 1.9 kg.




## 2.3 UAF reference measurements

The CTC site was the ALPACA reference site for outdoor pollution in downtown Fairbanks (Simpson et al., 2024). At this site, trace gases (NO, $NO_2$, $O_3$, CO and $SO_2$) and $CO_2$, were measured by reference instruments from January $1^{st}$ to March 16 2022 (Cesler-Maloney et al., 2024). CO was measured with a gas filter correlation analyser, $O_3$ with an ultraviolet photometric analyser, NO and $NO_2$ with a chemiluminescence analyzer. The gas analysers were calibrated roughly weekly at the CTC site. The data were delivered with minute and hour averages. The CTC hourly averaged timeseries of temperature, difference

of temperature between 3 m and 23 m, humidity, $CO_2$ and trace gases concentrations are displayed in Figure 1. Temperature variations were important during the campaign with the occurence of alternating warm and cold periods. It is particularly apparent for the period from January 29 to February 10. The first part of this period (January 29 to February 3) is characterised by low temperatures and high surface pressures (not shown) corresponding to anticyclonic conditions (Fochesatto et al., 2024). Such conditions promote the formation of a SBI (Mayfield and Fochesatto, 2013). The temperature inversion is clearly seen

on Figure 1 with the enhanced 23-3 m temperature gradient. The trapping of pollutants in the SBI is also captured with hourly NO concentrations above 100 ppbv and enhanced CO (> 1 ppmv) and $CO_2$ concentrations. It was so polluted that titration by NO resulted in $O_3$ levels lower than 1 ppbv during that period. The anticyclonic period ends on February 3 during the early afternoon leading to an abrupt increase in temperature, the decline of the SBI, and the decrease of NO, $NO_2$, CO, $CO_2$ and increase of $O_3$. The temperature at 3 m rises by 10 ° C (from -27 to -17 ° C), the 23m-3m inversion drops from 5.5 to 0.8 ° C

and NO from 167.8 to 5.5 ppbv within only 4 hours (from 11AM to 3PM).

The trace gas concentrations at CTC are therefore highly variable depending on the local meteorological conditions. This was taken into account when choosing the learning and prediction datasets, notably the concentration ranges, to perform and validate the calibration of the EGSs.

## 2.4 MICROMEGAS deployement strategy


During the ALPACA 2022 field experiment, the MICROMEGAS instrument was deployed in 3 different ways. It was operated at the CTC site for calibration against reference analysers for 7 periods corresponding to a total of 250 hours (see Figure 2). At CTC, the temperature recorded on one of the EGSs (see Figure 3) displays a moderate variability (11 to 28 °C) compared to the outside temperature which varies from -34 to 5 °C (see Figure 1). According to their datasheets, the A4

sensors can be operated over a wide range of temperatures, from -30 to 50°C for NO, $NO_2$ and CO and -20 to 50°C for $O_x$ (www.alphasense.com). The thermal regulation of the sensors limits the effect of outside temperature variations on the measured concentrations. Nonetheless, this effect is accounted for calibrating the data as described in section 2.5. The ouside air RH sampled by the MICROMEGAS SHT75 sensor varies between 28 and 83% with the lowest values recorded during the coldest period from January 29 to February 3.

MICROMEGAS was also deployed 5 times between 21 January and 16 February in a vehicle for surface mapping of pollution in and around Fairbanks. The MICROMEGAS insulated box was placed inside a larger plastic box on the roof of the vehicle.

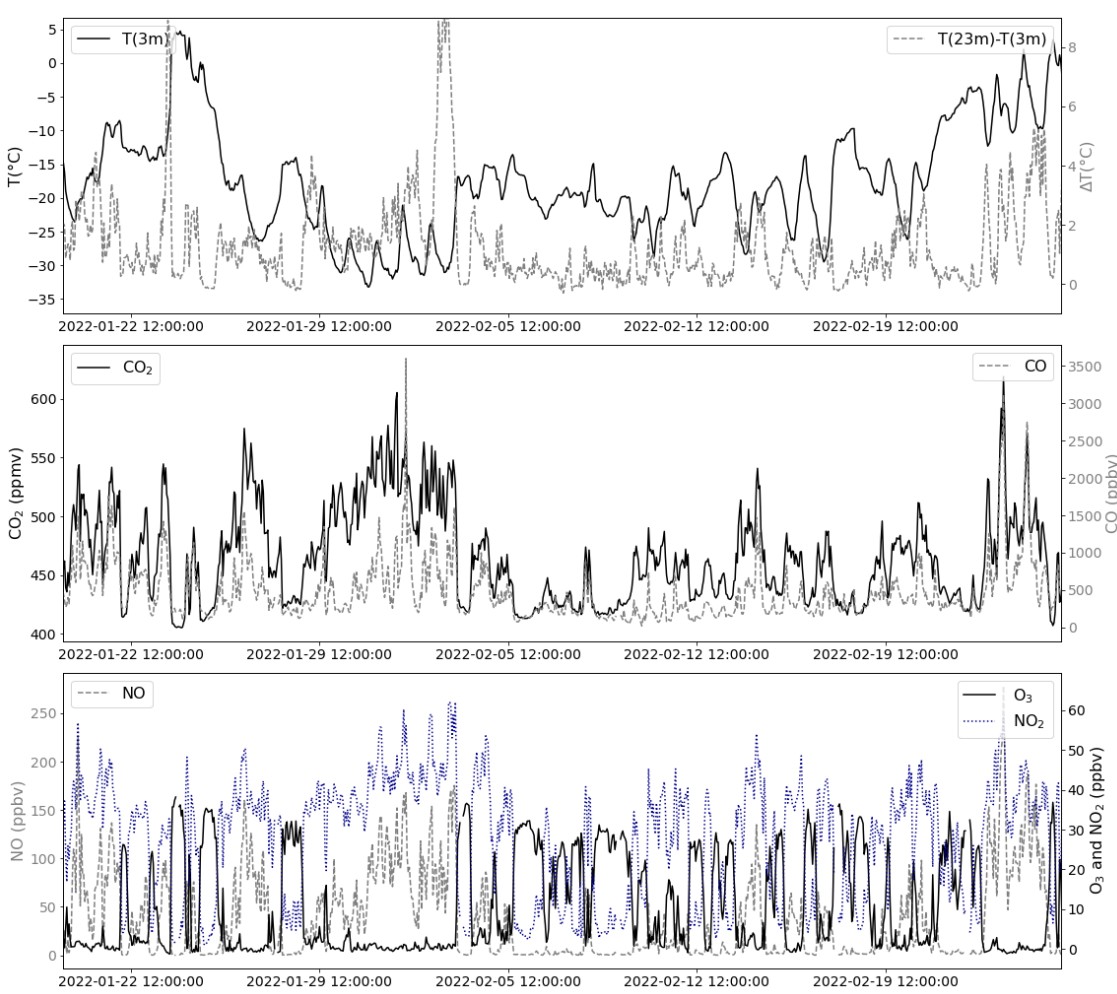

**Figure 1.** Time series of hourly averaged CTC measurements: (top) temperature at 3 m (red, left axis) and T(23m) - T(3m) (grey dashed line, right axis) (middle) $CO_2$ (black solid line, left axis) and CO (grey dashed line, right axis) (bottom) NO (grey dashed line, left axis), $NO_2$ (blue dotted line, right axis) and $O_3$ (black solid line, right axis).



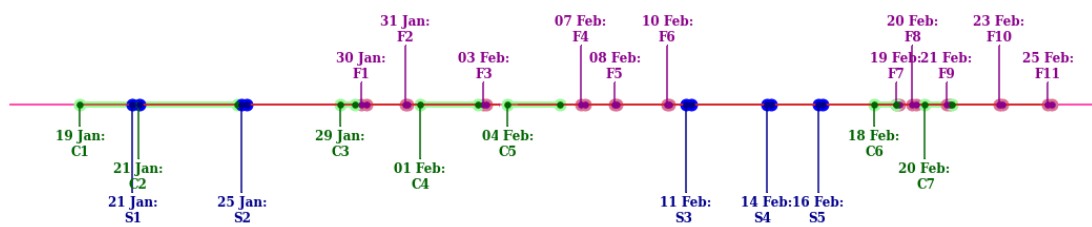

**Figure 2.** Timeline of MICROMEGAS operations during ALPACA 2022: (green, C1 to C7) calibration periods at the CTC site, (red, F1 to F11) flights onboard the Helikite at the UAF Farm site, (blue, S1 to S7) car ('Sniffer') rounds onboard road vehicles.

The vehicle was driven at a speed of less than 20 m.p.h. in order to sample distances of less than 500 meters in one minute. It also stopped frequently for a few minutes to sample specific locations.

Finally, MICROMEGAS was incorporated in the MoMuCAMS platform to be deployed with other instruments onboard the Helikite at the UAF-Farm site. The balloon payload was designed to sample vertical profiles of trace gases and particles and possibly intercept pollution plumes in the ABL. The MICROMEGAS instrument weighs only 2 kg. It could therefore replace the MoMuCAMS CO and $O_3$ instruments for less than half their weights to save space and weight in order to fly with a more complete aerosol package (see Pohorsky et al. (2024b)). More importantly, it also measures NO and $NO_2$ not part of the Mo-

MuCAMS instrumental package. MICROMEGAS performed 11 successfull balloon flights. Nighttime flights reached higher altitudes (up to 350 m a.g.l.) and lasted longer (up to 5 hours) than daytime flights because of air traffic regulations.

Data from the MoMuCAMS CO and $O_3$ instruments were compared to the calibrated MICROMEGAS observations when the instruments were jointly operated at the UAF-Farm between Helikite flights (section 3.2).

**2.5 Calibration methodology**

As mentioned in the introduction, low-cost EGSs calibration can be conducted under controlled atmospheric conditions (gas concentrations, temperature, humidity) in laboratories or ambient conditions in the field. The latter option allows the sampling of more realistic conditions that better match the environmental conditions encountered during observations. Furthermore, during the ALPACA-2022 campaign, we had observations with reference analyzers for five target gases (CO, $O_3$, NO, $NO_2$ and

$SO_2$) at the CTC site. We have therefore chosen the second option for our calibration.

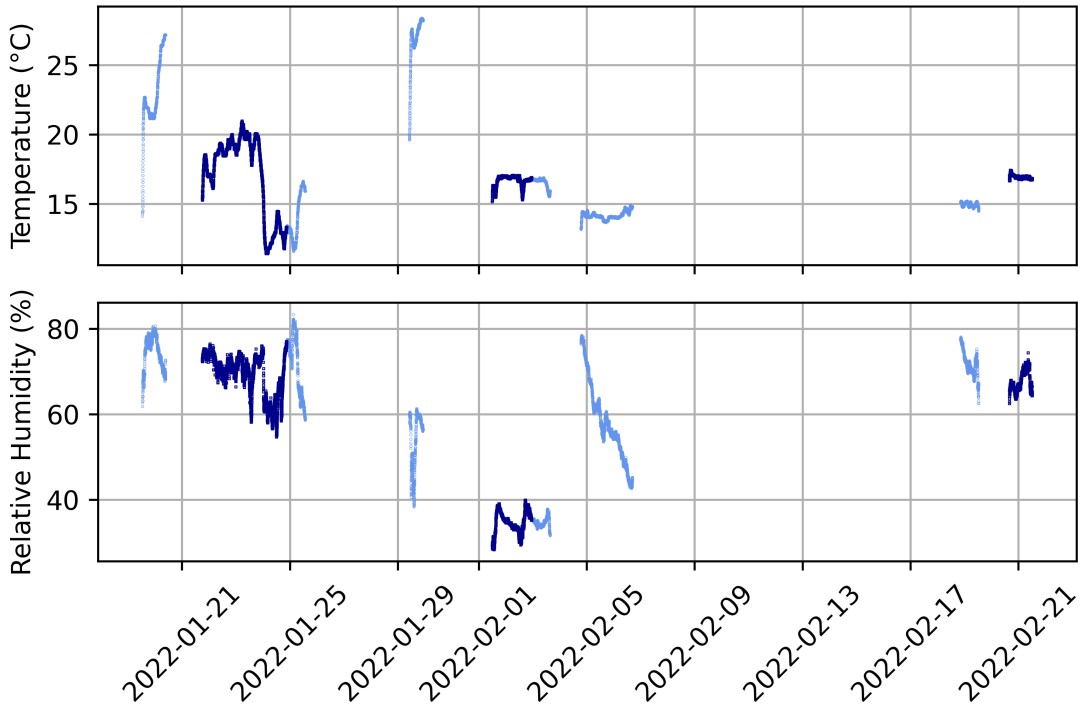

**Figure 3.** Time series of: (top) Temperature of EGSs for learning (light blue) and prediction (dark blue) periods; (bottom) Relative Humidity (RH) of air sampled by the EGSs.

### 2.5.1 Regression method

The calibration methods themselves fall into two categories: parametric and non-parametric methods. Many studies use parametric ML methods that simplify the function to be fit to a known form. The simplest form are univariate or multivariate linear or quadratic functions which are often used as reference methods (Spinelle et al., 2015, 2017; Bigi et al., 2018; Casey et al., 2019; Malings et al., 2019; Liang et al., 2021; Bittner et al., 2022). More sophisticated forms could be Artificial Neural Networks (ANN, Casey et al. (2019); Malings et al. (2019); Spinelle et al. (2015, 2017)). Non parametric ML algorithms have also been used such as decision trees (Bigi et al., 2018; Smith et al., 2019), support vector machines (Bigi et al., 2018), gaussian processes (Smith et al., 2019; Malings et al., 2019) or k-nearest neighbor clustering (Malings et al., 2019; Bittner et al., 2022). To take advantage of parametric and non parametric methods, some authors have also developed hybrid models combining for instance linear regression and random forest (Zauli-Sajani et al., 2021; Zimmerman et al., 2018; Malings et al., 2019; Bittner et al., 2022).

Both parametric and non-parametric methods have their advantages and drawbacks. Non-parametric methods generally provide better fits than parametric methods because the form of the relationship between the EGS gas concentrations (output) and input





parameters (working and auxiliary electrode voltages, humidity and temperature) is less constrained. Nevertheless, they have difficulties for prediction in conditions (target or interfering gas concentrations, temperature or humidity) that fall outside the learning dataset. In contrast, parametric models generally allow broader and more reliable extrapolations to conditions outside those of the learning dataset. However, they are subject to a lower flexibility of the input-output relationships which are set to a given function or set of functions.

Spinelle et al. (2015) (resp. Spinelle et al. (2017)) proposed linear regressions (LR) or multivariate linear regressions (MLR) and ANN methods for the calibration of $O_3$ and $NO_2$ (resp. NO, CO and $CO_2$) sensors. As they concluded that simple LR and MLR are characterized by high uncertainties, we tested MLR and slightly different linear methods. LR minimizes a difference in order to reproduce best a mean value. Quantile regressions (QR) allow to minimize the differences to reproduce best a given quantile (q) such as the median (q = 0.5). We therefore tested QR methods with q=0.25 (QR0.25), 0.5 (QR0.5) and
0.75(QR0.75).

Spinelle et al. (2015) also used two types of ANN, radial based functions and multi-layer perceptron (MLP). They found out that the former did not yield satisfactory results and thus discarded it. Spinelle et al. (2015) and Spinelle et al. (2017) relied on MLPs, because they provided very good results. Following their recommendations, we tested calibrations with MLP. MLP is a supervised learning algorithm that requires tuning multiple hyperparameters to achieve optimal results. After preliminary tests
varying these hyperparameters, the results of the MLP calibration were not found to be very sensitive to the number of neurons, layers and iterations that we set to 10, 10 and 4000 respectively. The results appeared more sensitive to the regularisation parameter $\alpha$ (introduced to mitigate overfitting). Large $\alpha$ values promote smaller weights improving the fit for high variances but potentially leading to overfitting; lower $\alpha$ values favour larger weights, and, hence, better fix high biases which may result in underfitting. Based on our preliminary tests we present the best results which were achieved with $\alpha$ between 1 and 1000.

Non parametric methods such as Random Forest (RF) are also used for EGS calibration (Bigi et al., 2018; Smith et al., 2019). We tested two renowned ensemble non parametric methods, the Histogram-Based Gradient Boosting Trees (HGBT) and the RF. Our calibration ML tools are based on Python libraries from the scikit-learn initiative (https://scikit-learn.org/) described in Pedregosa et al. (2011). The calibration data were split into two equal and independent parts of ~125 hours of joint MICROMEGAS and reference analyser observations at CTC (see Figure 3); the first part was used for the learning of
the calibration functions (Equation (1)), and the second part of the data was used to validate the predictions made by these functions. Learning and prediction datasets are chosen to contain both very cold and highly polluted periods and warmer and less polluted periods. The pollution levels of the learning and prediction periods can clearly be inferred from NO levels on Figure 7. It has to be noted that the learning and prediction periods were identicals for the 4 gases.

## 2.5.2 Regression parameters

As mentioned earlier, the output voltages of the EGS depend not only on the concentration of the target gas, but also on that of interfering gases, as well as **RH** and temperature (**T**). Therefore the calibration function $f$ to be adjusted must be a function of all these parameters. As we do not have direct measurements of the interfering gases, we use the voltage of the working





electrode $V_w$ and of the auxiliary electrode $V_a$ of the EGS targeting these gases when available. For the EGS targeting gas **G0**
with cross-sensitivities to gases **G(i)** with i = 1 to n, the generic calibration function can be written as:

$$[\textbf{G0}] = f(V_\text{w}(\textbf{G0}), V_\text{a}(\textbf{G0}), V_\text{w}(\textbf{G(1)}), V_\text{a}(\textbf{G(1)}), ..., V_\text{w}(\textbf{G(n)}), V_\text{a}(\textbf{G(n)}), \textbf{RH}, \textbf{T}) \tag{1}$$

According to its datasheet from Alphasense, the CO-A4 EGS does not present important cross-sensitivities to other trace
gases. Furthermore, Liang et al. (2021) do not use interfering gases for the calibration of CO-B4 Alphasense sensors in moni-
toring AQ in Chinese cities. Therefore, no interfering gases are introduced in the calibration function (equation 2.5.2) for CO.
With laboratory measurements, Lewis et al. (2016) showed that for Alphasense B4 EGS the working electrode of $O_x$ sensors
is sensitive equally to $O_3$ and $NO_2$, the electrode of NO sensors to NO and $NO_2$ and the electrode of $NO_2$ sensors mostly to
$NO_2$ and very little to NO. Some regression models for NO and $NO_2$ include the net ($V_w$ - $V_a$) voltages from the NO and
$NO_2$ sensors (Bigi et al., 2018). Based on these studies with varying approaches, we performed sensitivity tests with various
combinations of trace gases for the NO, $NO_2$ and $O_x$ sensors. For NO, the addition of $NO_2$ voltage as variable in its calibra-
tion function makes no significant difference and no interfering gases are accounted for in the NO regression. For $NO_2$, the
addition of voltages from the NO sensor in equ. 2.5.2 slightly improves the agreement with the reference data and therefore
NO is accounted for in the $NO_2$ regression as a cross sensitive gas. For $O_x$ the best results are obtained with the addition of the
voltages of the $NO_2$ and also of the NO sensor. The concentration of $O_3$ is then obtained by substracting [$NO_2$] from [$O_x$].

### 2.5.3 Evaluation statistics

To choose the best calibration method for each trace gas, it is necessary to evaluate how the fit obtained with each method re-
produces the reference data in terms of absolute value (accuracy) and variability (precision). The accuracy (systematic error), is
quantified by the Mean Bias Error (MBE) (i.e. average difference between the reference and the calibrated data); the precision
or average magnitude of the errors is approximated by the Root Mean Square Error (RMSE) (i.e. the square root of the average
of the squared differences between reference and calibrated data). The agreement regarding the phase of the variations can be
assesssed with the correlation coefficient R. The agreement regarding the amplitude of the variations can be evaluated with the
ratio between the standard deviation of the calibrated data and that of the reference data.

The Taylor diagram, commonly used in meteorology and climate science, takes advantage of the trigonometric relationship
between RMSE, standard deviations and correlation coefficients to display synthetically the performances of multiple datasets
against a reference dataset (Taylor, 2001). It consists of a circular grid with each dataset represented by a point and the reference
dataset placed at the center of the X-axis (e.g. see Figure 4). The distance between a data point and the reference point povides
the RMSE of the experiment and the distance from the centre of the diagram provides the ratio of the standard deviations.
The correlation coefficient between the reference and the experiment is given by the azimuthal position of the point. For each
experiment, the RMSEs and standard deviations are normalised by the standard deviation of the reference data to display the





| Species | Method | Learning | | | Prediction | | |
|---|---|---|---|---|---|---|---|
| | | R | MBE±RMSE (ppbv) | slope | R | MBE±RMSE (ppbv) | slope |
| CO | raw | 0.86 | $14 \pm 186$ | 1.24 | 0.82 | $-7 \pm 290$ | 1.12 |
| NO | MLP 100 | 0.99 | $0 \pm 7$ | 0.99 | 0.97 | $3 \pm 12$ | 1.04 |
| $NO_2$ | MLP 100 | 0.98 | $0 \pm 3$ | 0.98 | 0.98 | $1 \pm 3$ | 1.00 |
| $O_3$ | MLP 100 | 0.98 | $0 \pm 2$ | 0.98 | 0.95 | $0 \pm 4$ | 0.93 |

**Table 1.** Statistics of the comparisons between MICROMEGAS and reference data at CTC: correlation coefficients (R), mean bias error (MBE), root mean square error (RMSE) and slope of the regression line fitted (MICROMEGAS versus reference) between both datasets.

results from multiple experiments on a single diagram.

# 3 Results

## 3.1 Validation against reference measurements

For comparison with reference data at the CTC site, MICROMEGAS data were averaged in 1-minute intervals. For the 4 trace gases the correlation coefficient R, MBE and RMSE from comparisons between MICROMEGAS calibrated data and reference learning and prediction data at CTC are gathered in Table 1 for the selected calibration methods. We also provide the slope of the linear regressions fitted between MICROMEGAS and reference data.

The choice of the calibration method for each target trace gas is explained in the following subsections which present the detailed calibration results for CO (section 3.1.1), NO (section 3.1.2), $NO_2$ (section 3.1.3) and $O_3$ (section 3.1.4).

### 3.1.1 CO

The Taylor diagram of the CO sensor displays the results from the different calibration methods for the learning and prediction
datasets at CTC (Figure 4(a)). As expected, the performances are better for the learning than for the prediction dataset with larger correlation coefficients (0.86<R<0.96 for learning and 0.74<R<0.83 for prediction) and ratios of variabilities relative to the reference data closer to unity. The HGBT and RF achieve the best agreement for learning but almost the worst for prediction. These non-parametric methods are characterised by a strong flexibility that enable them to match the reference dataset very well but have difficulties predicting data that fall even slightly outside of their learning database. In our case both learning
and prediction datasets are chosen within periods with similar weather and pollution conditions but with some differences that are probably the reason for the lower performances with the prediction dataset.

Except for the non-parametric methods, all the calibration methods have R values slightly larger than 0.8 for the prediction



dataset (Figure 4(a)). The differences come from their ability to reproduce the amplitude of the CO variability from the reference dataset with ratios varying from 0.71 (MLP1.0) to 0.97 (QR0.75). The performances of the raw data in reproducing the variability of the reference data are almost similar to the performances of QR0.75 but the raw data MBE is much smaller, -7 instead of 61 ppbv (see Table 1) and we have therefore chosen to use the raw data for the CO EGS.

The time series of CO measurements (raw data) from MICROMEGAS and the reference CO analyser are displayed in Figure 5. MICROMEGAS captures well the very large CO variations from hundreds to thousands of ppbv. However, the absolute biases can reach 1 ppmv for the highest concentrations. As a result, the absolute RMSE between the reference and MICROMEGAS CO data for the prediction dataset is rather large (290 ppbv).

### 3.1.2 NO

The NO Taylor diagram (Figure 6(a)) differs greatly from the CO one. All the experiments have correlation coefficients R>0.9, and the majority of them are even larger than 0.95. The variabilities in the MICROMEGAS experiments are mostly in the range 0.90-1.05 times the variabilities of the reference data and the RMSEs are lower than 30% of the reference data variability. We have chosen the MLP100 method because it displays the largest R and a variability ratio of 1.0. Furthermore, the slope of the linear regression between reference and MICROMEGAS data is very close to unity (Figure 6(b) and Table 1) with only a very limited number of negative MICROMEGAS NO values which do not exceed a couple of ppbv (Figure 6(b)).

The ability of MICROMEGAS to capture the NO large variability (0-250 ppbv) is illustrated by the time series at CTC displayed on Figure 7. The discrepancies are larger for the prediction than for the learning dataset with biases reaching ±50 ppbv and ±30 ppbv respectively. Nevertheless, the global RMSE for prediction remains moderate at 12 ppbv and the MBE of 3 ppbv is not significant.

### 3.1.3 NO$_2$

The NO$_2$ Taylor diagram is very similar to the NO one (Figure 8(a)). Most experiments have R>0.95 and variability ratios are even closer to unity than for NO. For the prediction, most experiments provide very similar results. We have chosen MLP100 which performs slightly better than MLP1 or MLP10. Even if they are able to reproduce the variability of the reference data correctly, linear regressions are excluded because, contrary to MLP, they provide significantly negative values. The raw calibration method displays a good correlation with the reference data but only just half of its variability.

The scatter plot between MICROMEGAS and reference NO$_2$ data (Figure 8(b)) shows excellent agreement with a the linear regression slope of unity.



As expected from the previous analysis, MICROMEGAS data follow closely the NO₂ variations from the reference analyser (Figure 9). The difference between both datasets is within $\pm 10$ ppbv ($95^{th}$ percentile) with a very low mean bias (1.5 ppbv) and a RMSE (3 ppbv) about 4 times lower than for NO.

### 3.1.4 O₃

Once $O_x$ and $NO_2$ are calibrated, we compute $O_3 = O_x$ - $NO_2$. For $O_3$, the prediction results are characterised by decreased performances relative to the learning results and to the $NO_2$ prediction results (Figure 10(a)). The best calibration method is clearly MLP100 with a correlation coefficient of 0.95 and an amplitude of variability only 10% lower than the reference's one. The scatter plot displays the bimodal $O_3$ distribution with low values (< 5 ppbv) from almost complete $O_3$ titration by NO during high pollution periods, and $O_3$ between 20 and 40 ppbv during cleaner periods. The linear regression slope (0.93) is

lower than for the other gases (Table 1) but remains quite close to unity. The MLP100-calibrated $O_3$ data are never negative contrary to $O_3$ from linear calibration methods.

MICROMEGAS $O_3$ captures the variations of reference $O_3$ (Figure 11) with alternance of polluted periods with little $O_3$ and cleaner periods with about 30 ppbv of $O_3$ in anti-correlation with NO levels (Figure 7). For the prediction dataset, biases

can reach absolute values larger than 10 ppbv for highest levels of $O_3$ but the mean bias is 0$\pm$4 ppbv.

### 3.2 Comparisons of CO and O₃ from MICROMEGAS to analysers from MoMuCAMS at the UAF-Farm site

MICROMEGAS was operated on the ground at the UAF-Farm site before the Helikite flights or between two successive flights for five periods of hours to days. When they were not flying, both CO and $O_3$ MoMuCAMS instruments (see section 2) were also operated at the UAF-Farm site. The MoMuCAMS CO PICO (resp. $O_3$ 2B Tech) instrument was operated for 129 (resp.

105) hours in coincidence with MICROMEGAS at the site. The results from comparisons between the CO and $O_3$ instruments are displayed in Figure 12 and 13 respectively.

The UAF-Farm site at the edge of the city is less polluted than the CTC site downtown. The CO concentrations remain within the 100-300 ppbv range (Figure 12) while CO concentrations at CTC are mostly over 500 ppbv and often exceed 1 ppmv (see Figure 1). At the UAF-Farm, $O_3$ is seldom fully titrated by NO with concentrations mostly between 20 and 40 ppbv and below

10 ppbv over rare and short periods (Figure 13). It is in contrast to CTC where $O_3$ is fully titrated by NO over several periods that can last many days (Figure 1).

CO from MICROMEGAS has no systematic bias (-2 $\pm$ 50 ppbv) relative to the PICO instrument. The RMSE (50 ppbv) is much lower than at CTC because the absolute CO values are much lower. According to Figure 12, the absolute differences between both instruments rarely exceeds 50 ppbv. The correlation coefficient (R = 0.81) between MICROMEGAS and PICO

CO data at UAF-Farm is very close to the one for the CO prediction data at CTC (Table 1).



For $O_3$, no systematic bias ($0 \pm 4$ ppbv) is observed between MICROMEGAS and the 2B-Tech instrument. Moreover, the MBE (0 ppbv) and RMSE (4 ppbv) are identical to those computed for the prediction dataset at CTC (Table 1). The correlation coefficient (R = 0.86) is nonetheless lower than at CTC (Table 1). This is probably related to the $O_3$ variability which is lower at the UAF-Farm with no complete $O_3$ titration and only few periods with low $O_3$ concentrations.

### 3.3 On-road mobile sampling

As mentioned in section 2, MICROMEGAS was deployed 7 times on the roof of a vehicle to perform on-road mobile samplings in Fairbanks and its surroundings. We present here MICROMEGAS measurements from the first drive performed on January 21 from 12:39 to 16:40 PM. The car was parked in a street next to the CTC measurement site at the start (12:39 to 13:09) and at the end (16:32 to 16:38) of the drive. The car first headed towards the residential neighborhood of Hamilton Acres and stoped next to the instrumented site The House. The next top was at the Birch Hill Recreation Area. The car then headed west on College Road and reached the UAF-Farm site after a stop at the Aurora residential area and drove up to the top of Chena Ridge via Chena Ridge Road before descending towards Fairbanks Airport via Chena Pump Road. It then headed east on Parks Highway to reach the CTC site by going north (see Figure 14).

According to CTC data (see Figure 1) the period of the drive is characterized by relatively warm temperatures for the season ($\sim$ -10°C) and significant temperature inversions between 3 and 23 m altitude (between -2 and -1.5°C). It is identified as strongly stable according to Brett et al. (2024) promoting elevated pollution levels. During the 4 hours of the drive, the pollution levels increased at CTC from 590 to 1610 ppbv for CO and from 40 to 110 ppbv for NO (see Figure 15). Such increases probably result from the diurnal variability of road traffic.

The comparison between the CTC reference measurements and MICROMEGAS data (Figure 15) clearly indicates that MICROMEGAS captures the variations of trace gas concentrations between the start and the end of the drive. It is noteworthy that MICROMEGAS is also able to capture the CO and NO variability very well for the half hour at the CTC site before the car moves away. For $O_3$ the concentrations are close to zero during both stops at CTC and the agreement is excellent with biases lower than 0.2 ppbv. MICROMEGAS slightly overestimates $NO_2$ (5.4 ppbv) at the beginning and the bias decreases to 0.1 ppbv at the end. The NO bias is negligible at the start (-0.3 ppbv) and MICROMEGAS is underestimating NO by 14 ppbv at the end. For CO, the biases are larger at the beginning (210 ppbv) than at the end (101 ppbv). For the four trace gases, the biases are in good agreement with the intervals from Table 1. It has to be noted that at the beginning MICROMEGAS sampled a peak with largely enhanced CO (and slightly enhanced NO) which was not detected by the reference analyser (Figure 15). This discrepancy is most likely resulting from the location of MICROMEGAS on the roof of the car which was parked about 10 meters away from the CTC trailer and closer to the traffic emissions than the analysers inlet located on the roof of the trailer. Therefore, MICROMEGAS probably sampled a plume from the traffic that did not reach the analysers inlet.

The pollutant maps of Figure 14 display significant variations among the sampled areas. The first striking feature is that the $O_3$ concentrations (Figure 14 (c)) are strongly correlated with the altitude (Figure 14 (a)) on both uphill legs of the drive





reaching Birch Hill to the North East and Chena Ridge to the South West. Elevated areas are indeed isolated from the strong emissions from the city, especially during temperature inversion periods. The background air sampled on the hill slopes is therefore also characterised by CO and NOx concentrations lower than in the city. The large concentrations of NOx and CO on the south leg of the drive from the airport to CTC following Airport way, Parks and Richardson Highway, are due to sampling

of air impacted by the traffic during the rush hours. NOx concentrations are lower on the North leg on Johansen Expressway and College road probably because of a lower traffic at the beginning of the afternoon.

The two residential areas sampled at one hour intervall have different levels of pollutants. The Hamilton Acres area has a low level of NO (5 ppbv) and CO (450 ppbv) compared to the Aurora area where NO reaches 77 ppbv and CO 730 ppbv (see Figure 15). The levels of $NO_2$ are also larger at Aurora (47 ppbv) than at Hamilton Acres (35 ppbv). During the drive, at the beginning

of the afternoon little traffic was seen in those residential areas. Nevertheless, at the Aurora site, the 0-10 m winds simulated by the WRF model (see Brett et al. (2024)) are weak and blowing from the North bringing air polluted by traffic on Johansen Highway to the residential area to the South. Stronger winds are blowing from the North east (Figure 15 (a)) bringing clean air from outside of the city to Hamilton Acres. The differences in wind strength and direction probably explain the difference of pollution levels between the two residential areas. The area around the UAF-Farm to the Noth west of the Airport is marked

by enhanced $O_3$ and low NOx and CO concentrations as a result of the North flow bringing background air (Figure 14 (a)).

### 3.4 Vertical profiles from Helikite flights

As mentioned earlier (Figure 2), MICROMEGAS performed 11 Helikite flights in the MoMuCAMS platform from January 30 to February 25 2022. The Helikite was operated as follows: the balloon made an initial ascent at maximum vertical speed

(20 m/mn$^{-1}$) before descending in stages. Then, the cycle of ascents and descents is repeated several times. When a plume is detected ($CO_2$ and particles concentrations enhanced with respect to background concentrations), the balloon ascends and descends within the altitude range corresponding to this plume and make prolonged stops of a few minutes around the altitude of maximum concentrations (see Pohorsky et al. (2024a) for details). For the flights, MICROMEGAS data were averaged in 15 seconds bins corresponding to 5 m vertical displacement at maximum velocity.

The general statistics with the median values, the 95th, 80th, 20th and 5th percentiles computed over the 11 flights are displayed in Figure 16; Table 2 provides the median values, $20^{th}$ and $80^{th}$ percentiles computed over the whole dataset. For the four trace gases, the median values remain within relatively narrow ranges of values over the 0-350 m vertical range. Extreme values are particularly noticeable for $NO_2$ with peaks of up to 40 ppbv, ten times larger than the median value and for $O_3$ with troughs down to 5 ppbv. Nevertheless, extreme values of CO, NO and $NO_2$ observed during the flights remain low compared

to what was obeserved at CTC during long polluted episodes (Figure 1). This is due to the location of the UAF-Farm, which is far from the large pollution sources and rather upwind of the dominant winds from the northern sector at Fairbanks.

An example of a Helikite flight with a plume detected on January 30 was given in Simpson et al. (2024). During that flight, peaks of $NO_2$ are recorded between 50 and 100 m and between 200 and 250 m in coincidence with $CO_2$ enhancements. In

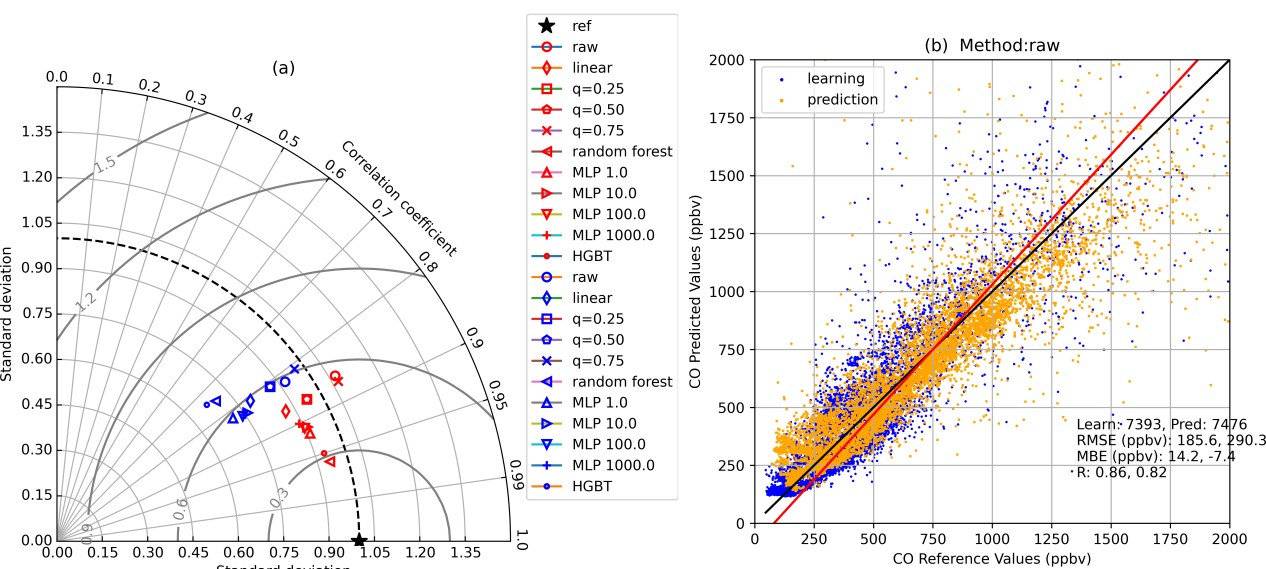

**Figure 4.** CO calibration methods: (a) Taylor diagram for MICROMEGAS versus CTC reference measurements: (red symbols) learning data (blue symbols) prediction data; (b) scatter plot of MICROMEGAS raw data versus CTC: (orange symbols) learning data (blue symbols) prediction data.

| Species | $20^{th}$ | median | $80^{th}$ |
|---------|-----------|--------|-----------|
| CO | 120 (134) | 140 (149) | 210 (179) |
| NO | -0.3 (-1.5) | 7.7 (-0.6) | 14.5 (0.8) |
| $NO_2$ | 1.5 (3.9) | 3.9 (6.7) | 9.8 (13.6) |
| $O_3$ | 26 (23.5) | 32 (30) | 38 (33.5) |

**Table 2.** Statistics ($20^{th}$ percentile, median and $80^{th}$ percentile) of the MICROMEGAS Helikite data for the whole ALPACA-2022 campaign and for the flight of February 20 (numbers between brackets).



Brett et al. (2024), MICROMEGAS $NO_x$ and CO flight data were used to evaluate Lagrangian simulations of pollution plumes originating from power plants to the east of the UAF Farm site and emitted aloft from elevated chimney stacks. They focused on the flights of 30 January, 8-9 and 25 February. Here, we present the data for the four trace gases (CO, NO, $NO_2$ and $O_3$) for the flight on the morning of February 20 when pollution plumes are detected aloft (Figure 17). The origin of the pollution plume and the layering of the ABL during this flight are discussed in a separate on-going study. We focus here on the multi species

measurements of such a plume with the MICROMEGAS calibrated data. The statistics over the whole flight are presented in Table 2. For CO, $NO_2$ and $O_3$, the median values are within the $20^{th}$-$80^{th}$ percentile ranges from the whole Helikite datasets. The NO values on the flight are generally lower than the median value; nonetheless, this difference is still consistent with the evaluation of NO made in section 3.1.2. Indeed, the mean bias between MICROMEGAS and CTC reference data (3±12 ppbv) encompasses the difference of -8.3 ppbv between the NO median of the whole Helikite dataset and of the February 20 flight.

The CO, $NO_2$ and $O_3$ values measured during the flight generally fall within the $20^{th}$-$80^{th}$ percentile from Table 2 but some extreme values are detected a few times at different altitudes. A plume located approximately between 120 and 180 m was sampled during three short periods (from 07:17 to 07:33, 09:54 to 10:11, and 10:46 to 10:59 Alaskan Time) during the flight. In the plume, CO and $NO_2$ concentrations reach maxima of respectively 417 and 46 ppbv and $O_3$ decreases to 0 ppbv following titration. NO is enhanced with a maximum value of 11 ppbv in the plume but only clearly between 09:55 and 09:58. Finally,

another plume is detected between 273 and 300 m from 09:22 to 09:33 with NO mixing ratios up to 27 ppb; it is correlated with $O_3$ titration but with very limited $NO_2$ and CO enhancements. According to the Lagrangian modelling of Brett et al. (2024), the plume at 150 m is originating from the UAF-C or Aurora power plant in Fairbanks and the 285 m plume is not attributed.

## 4   Conclusions

The MICROMEGAS instrument has been successfully operated at Fairbanks Alaska during the ALPACA field experiment in January-February 2022 to sample surface distributions and vertical profiles of trace gases in the wintertime Arctic Boundary Layer. MICROMEGAS includes EGSs to measure NO, $NO_2$, $O_3$, and CO, as well as temperature and relative humidity sensors, and a GPS for localisation. It has been deployed onboard a vehicle for 7 on-road sampling drives in Fairbanks and surroundings and at the UAF-Farm site to the north-west of Fairbanks onboard a tethered balloon (the Helikite) for 11 flights up to a

maximum altitude of 350 m a.g.l. In order to calibrate and validate the EGSs, we also operated MICROMEGAS coincident with reference analysers at the instrumented CTC site downtown in Fairbanks for ∼ 250 hours.

   For EGSs calibration, we have tested linear (quantile regressions) and non-linear (multi layer perceptron) parametric methods and also non-parametric (random forest and Histogram-Based Gradient Boosting Trees) methods. The use of Taylor diagrams

allowed us to evaluate the ability of the different calibration methods to reproduce the variability from the reference data and to determine the optimal method. For CO, the linear relationship between the gas concentration and the sensor voltages provided by the manufacturer (raw method) provides the best agreement for prediction data and we used no other calibration method. For



NO, NO$_2$ and O$_3$, an excellent agreement was reached for the learning data with the non-parametric random forest method but the performances are largely reduced for the predictions. The perfect learning is probably linked to the fact that non-parametric

methods, such as random forest, are able to fit a large number of functional forms contrary to parametric methods which have fixed functional forms. The degraded results for the prediction results from the lower ability of the non-parametric methods to extrapolate outside of the learning database relative to parametric methods. The multi-variate quantile regressions have correct performances concerning the reproduction of variabilities but provide too many negative values. This is due to the basic linear functional form that extrapolate to values which are outside of the reference data range. For NO, NO$_2$ and O$_3$, the best predic-

tion results are obtained with multi layer perceptron (MLP) artificial neural networks with 10 layers of 10 neurons when the network regularization parameter $\alpha$ is set to 100 (MLP100). The MLP parametric function is complex enough to reproduce the non linear relationships between the different parameters and allows extrapolation to reproduce data outside of the learning dataset which is essential for the use of the sensors away from the calibration site. For these three gases, the variability of the reference analyser is very well captured by the EGSs. The correlation coefficients (R) caracterising the agreement of the phase of the variations are ranging from 0.95 to 0.97 and the standard deviation ratios caracterising the agreement of the ampitude

of the variations are close to 1. The slopes of the linear regressions between the EGSs and reference data are very close to unity (0.93-1.12) for the 4 gases also indicating similar amplitude of the variabilities and no strong concentration dependences of the biases. The MLP also avoids obtaining values outside of the range of the reference dataset and especially negative values.

The selected calibration methods were applied to the MICROMEGAS data obtained during on-road mobile samplings in and around Fairbanks and Helikite flights at the UAF-Farm site. Comparisons between the MICROMEGAS data and the reference measurements at the CTC site at the beginning and at the end of the drive of January 21 confirmed the high quality of the calibrated data under challenging conditions, with the instrument on the roof of a car with very cold temperatures. Similarly, comparisons with surface CO and O$_3$ data recorded at the UAF-Farm site demonstrated that the sensors accurately reproduced

the variations of these gases with pollution conditions very different from those at the calibration site. The data from the drive of January 21 demonstrated the ability of the EGSs to capture the spatial variability of pollution in and around Fairbanks. In particular, the data highlighted the contrast not only between the surrounding hills, characterized by background concentrations, and the polluted city, but also between different residential areas and various traffic routes depending on the time of day. Flight data enabled the documentation of variations in trace gases with altitude in the ABL. The median concentrations of pollutants

are mostly representative of background conditions and display relatively weak vertical gradients, but significant variabilities were measured at the surface and at various altitudes. The variations in altitude are caused by pollution plumes from elevated sources such as power plants which were sampled at a few occasions when the UAF-Farm site was downwind of the city. The flight data from February 20 are exemplary of the ability of MICROMEGAS to quantify the trace gases concentrations in a power plant pollution plume.


Our study showed that EGSs provide a good solution for documenting the surface and vertical distributions of gaseous pollutants over large ranges of concentrations and under extreme cold conditions onboard mobile platforms. However, it should



be noted that, during ALPACA 2022, the balloon flights took place at a fixed site upwind of the main surface (on-road traffic, domestic heating) or altitude (power and heating plants) pollution sources, which significantly limited the sampling of pollu-

tion plumes. The deployment of such sensors on UAVs at various locations in and around polluted Arctic cities like Fairbanks would allow for better characterization of the dilution and physicochemical evolution of pollution plumes at different altitudes in th ABL. Regardless of the application of EGS, it is important to keep in mind that calibration with high-quality reference data is the crucial step for obtaining accurate measurements.

*Data availability.*  Final data presented in the study will be available to the scientific community two years after the conclusion of the study. Arctic-data.io (https://arcticdata.io/catalog/portals/ALPACA) provides a portal to archival repositories of the field study's data.

*Author contributions.*  B.B. is responsible for the development, calibration and operations of the MICROMEGAS instrument. He is the main author of the paper. P.M. designed the acquisition electronics of the MICROMEGAS intrument. N.B. participated to the onroad mobile samplings during the ALPACA field campaign and to the elaboration of the manuscript. R.P. and A.B. were responsible of the balloon and

MoMuCAMS platform operations and of surface meteorological and trace gases measurements at the UAF-Farm site. J.Schmale is the EPFL PI in charge of the helikite operations. K.S.L. is the PI of the French CASPA project and supervised French operations on the field during ALPACA. S.B., G.P., F.S., M.B. participated to the flight operations at the UAF-Farm site. G.J.F. was the PI of the UAF-Farm site operations and provided logistical and technical support for the balloon operations. W.S. is the ALPACA PI and responsible of UAF measurements at the CTC site. J.M. is co-I of the ALPACA field campaign. M.C.-M. operated and processed trace gases, temperature and $CO_2$ measurements

at the CTC site. S.R.A. participated to MICROMEGAS operations during the campaign. S.D. is the Italian CNR PI responsible for part of the Helikite instrumentation and surface measurements at the UAF-Farm. B.D'A. and J.S. are PIs of the French CASPA project. All co-authors participated to the elaboration and correction of the manuscript.

*Competing interests.*  At least one of the (co-)authors is a member of the editorial board of Atmospheric Measurement Techniques.

*Acknowledgements.*  We thank the entire ALPACA science team of researchers for designing the experiment, acquiring funding, making

measurements, and ongoing analysis of the results. The ALPACA project is organized as a part of the International Global Atmospheric Chemistry (IGAC) project under the Air Pollution in the Arctic: Climate, Environment and Societies (PACES) initiative with support from the International Arctic Science Committee (IASC), the National Science Foundation (NSF), and the National Oceanic and Atmospheric Administration (NOAA). We thank University of Alaska Fairbanks and the Geophysical Institute for logistical support, and we thank Fairbanks for welcoming and engaging with this research. K.S.L., B.D'A., J.S., B.B., P.M., S.B., N.B., acknowledge support from the Agence Na-

tional de Recherche (ANR) CASPA (Climate-relevant Aerosol Sources and Processes in the Arctic) project (grant no. ANR-21-CE01-0017), and the Institut polaire français Paul-Émile Victor (IPEV) (grant no. 1215) and CNRS-INSU programme LEFE (Les Enveloppes Fluides



et l'Environnement) ALPACA-France projects. S.R.A. acknowledges support from the UK Natural Environment Research Council (grant ref. NE/W00609X/1). G.J.F. acknowledges support from NSF grants 2117971, 2146929, and 2232282. R.P. and J.Schmale received funding from the Swiss National Science Foundation grant no. 200021-212101. S.D., G.P., F.S. acknowledge support from the PRA (Programma di Ricerche in Artico) 2019 programme (project A-PAW) and from the ENI-CNR Research Center Aldo Pontremoli. W.S. and M.C.-M. acknowledge support from NSF grants NNA-1927750 and AGS-2109134. J.M. acknowledges support from NSF grants NNA-1927750 and AGS-2029747.




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





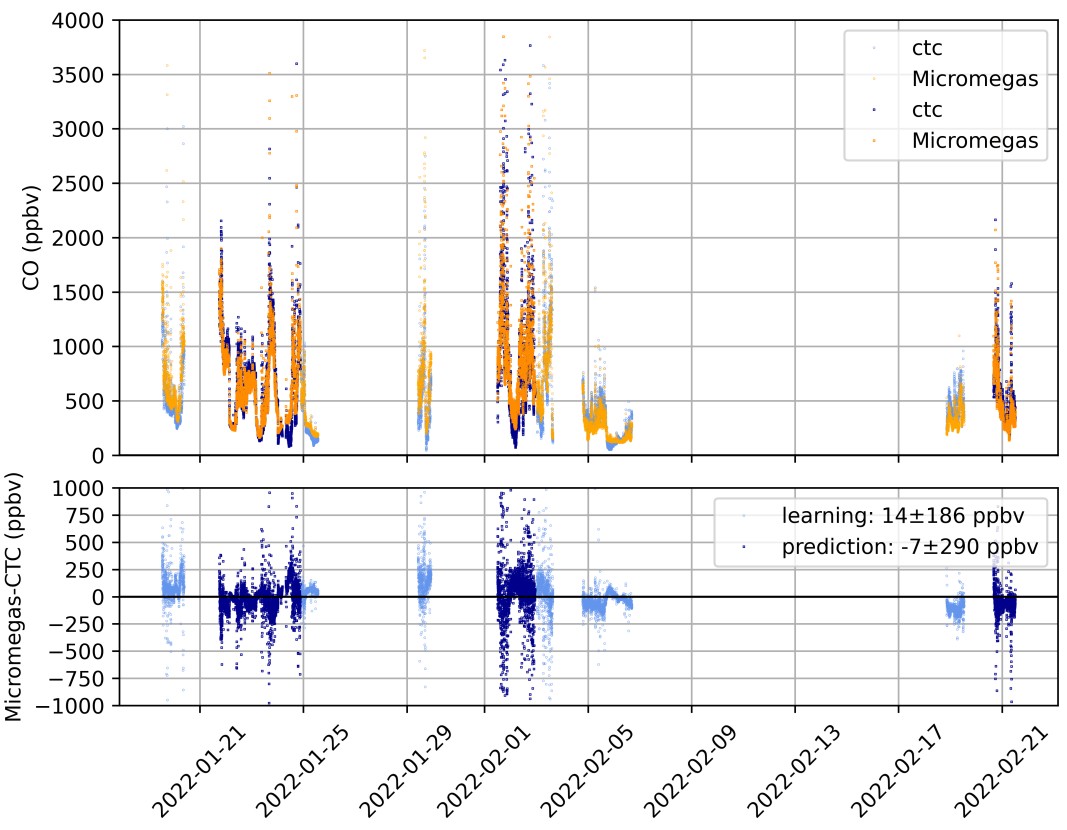

**Figure 5.** (top panel) time serie of CO measurements (raw data) at the CTC site (blue symbols) reference analyser data (orange symbols) MICROMEGAS data. Learning data in light colors and prediction data in dark colors. (bottom panel) time serie of the differences between the MICROMEGAS and the reference analyser data for (light blue symbols) learning data and (dark blue symbols) prediction data .





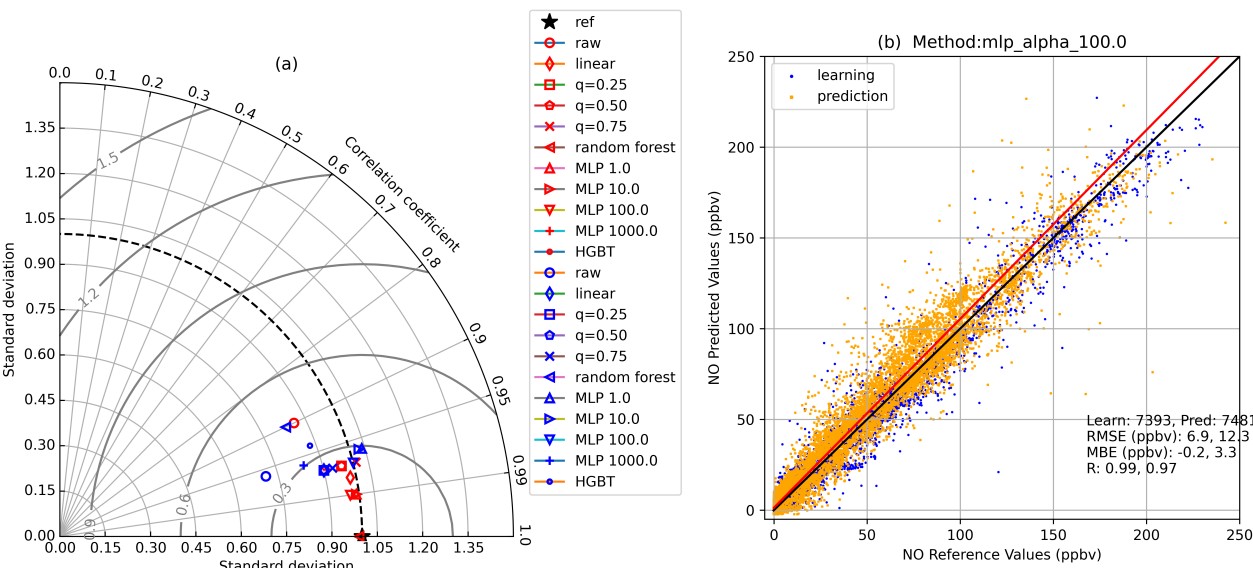

**Figure 6.** NO calibration methods: (a) Taylor diagram for MICROMEGAS versus CTC reference measurements: (red symbols) learning data (blue symbols) prediction data; (b) scatter plot of MICROMEGAS MLP-100 data versus CTC: (orange symbols) learning data (blue symbols) prediction data.



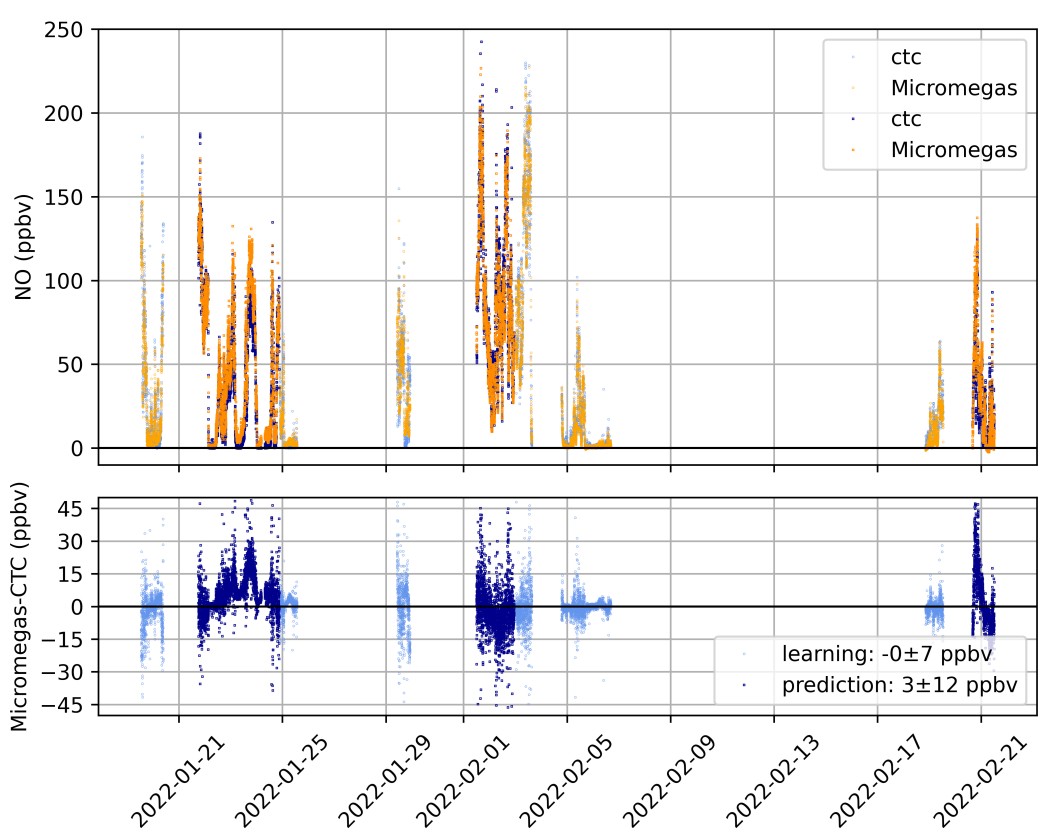

**Figure 7.** Same as Figure 5 for NO (MLP100 calibration function).



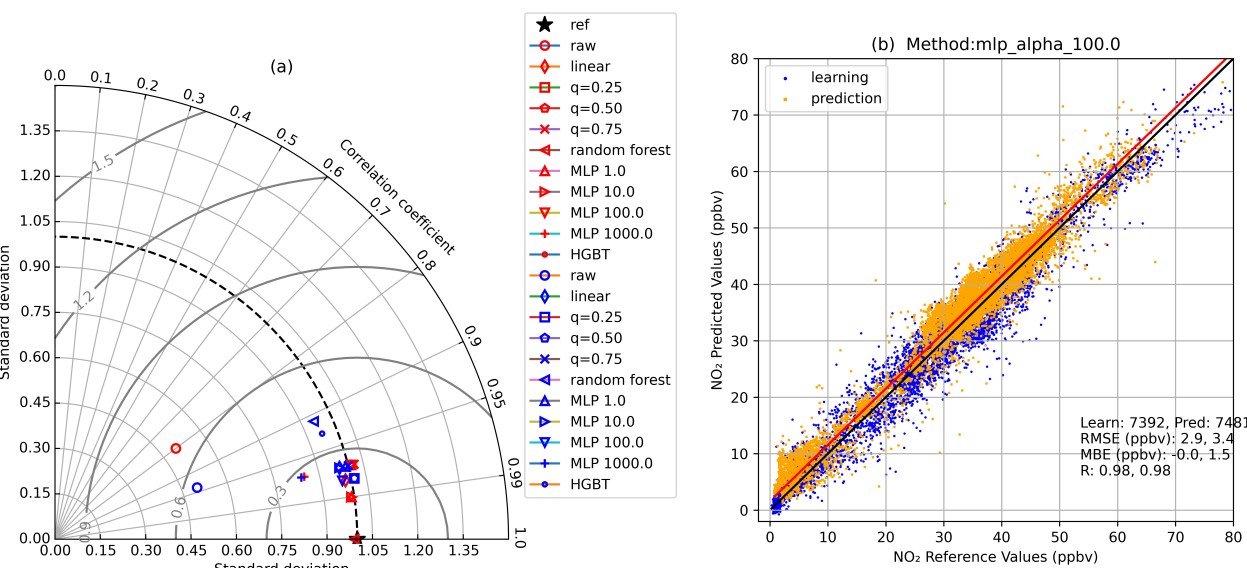

**Figure 8.** NO$_2$ calibration methods: (a) Taylor diagram for MICROMEGAS versus CTC reference measurements: (red symbols) learning data (blue symbols) prediction data; (b) scatter plot of MICROMEGAS MLP-100 data versus CTC: (orange symbols) learning data (blue symbols) prediction data.





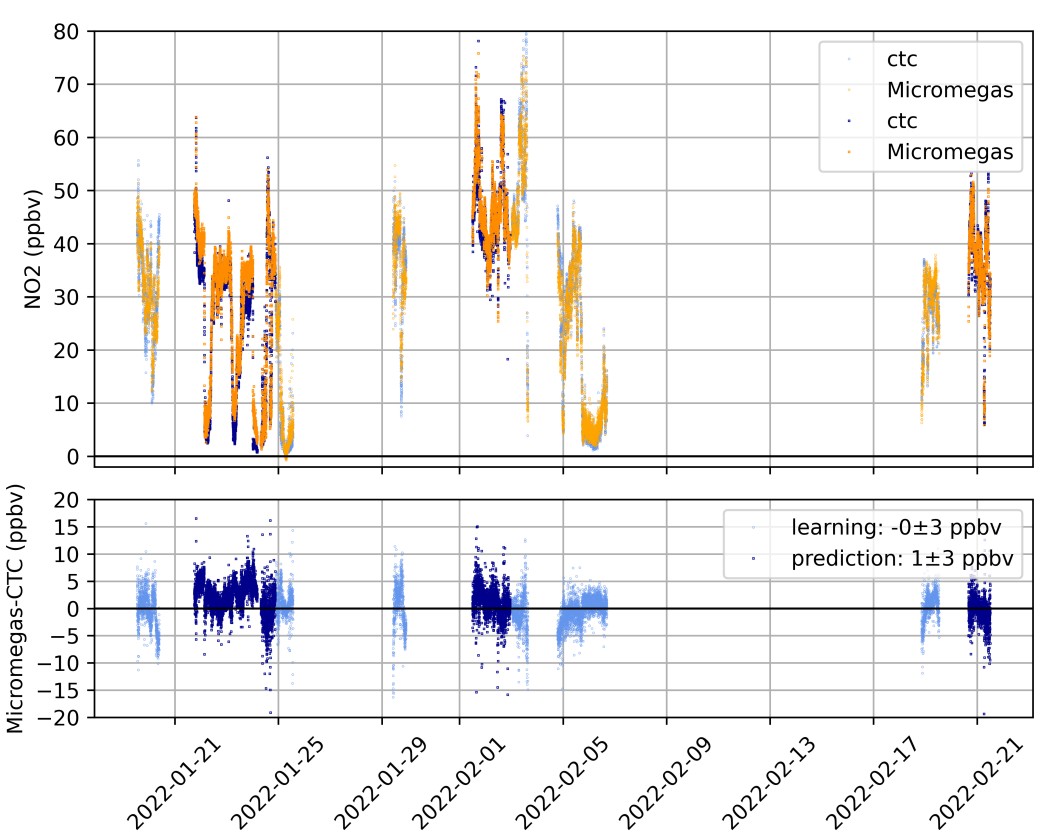

**Figure 9.** Same as Figure 5 for NO₂ (MLP100 calibration function) .





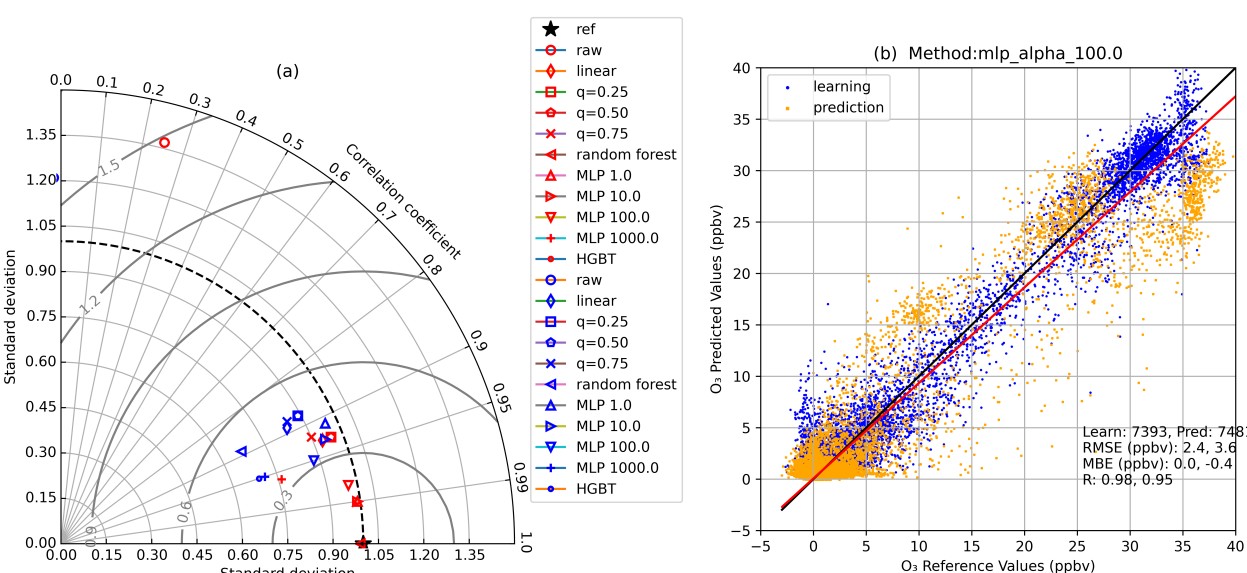

**Figure 10.** O$_3$ calibration methods: (a) Taylor diagram for MICROMEGAS versus CTC reference measurements: (red symbols) learning data (blue symbols) prediction data; (b) scatter plot of MICROMEGAS MLP-100 data versus CTC: (orange symbols) learning data (blue symbols) prediction data.



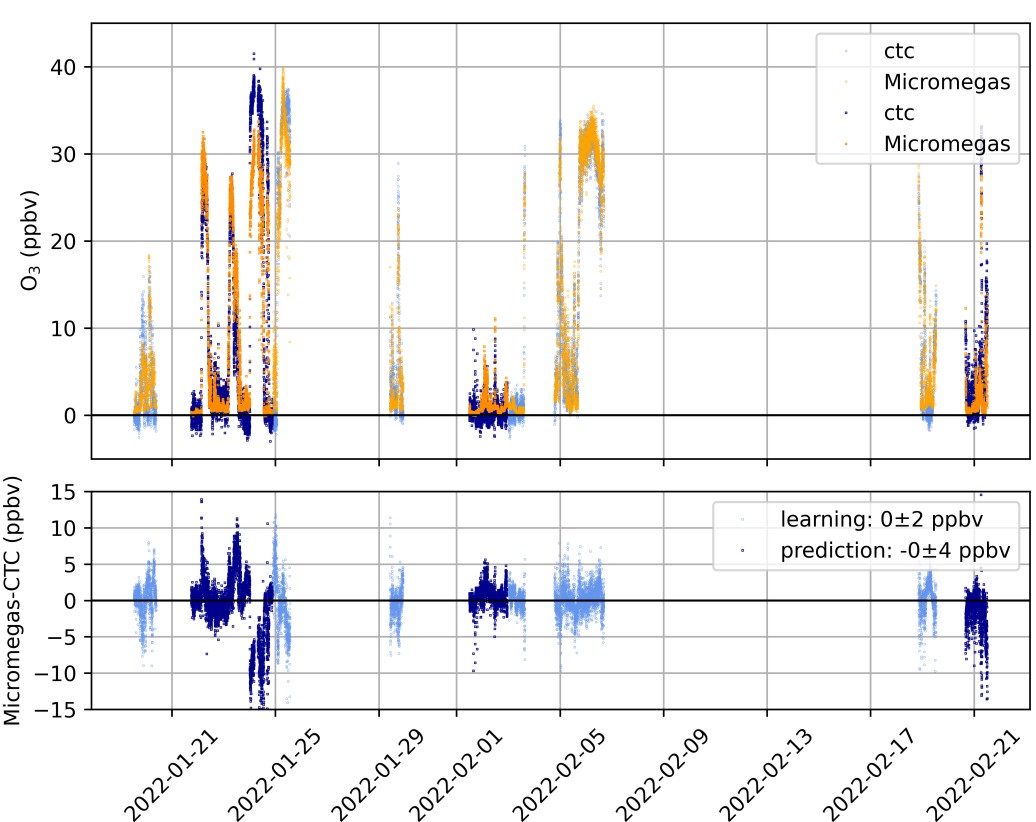

**Figure 11.** Same as Figure 5 for O$_3$ (MLP100 calibration function).





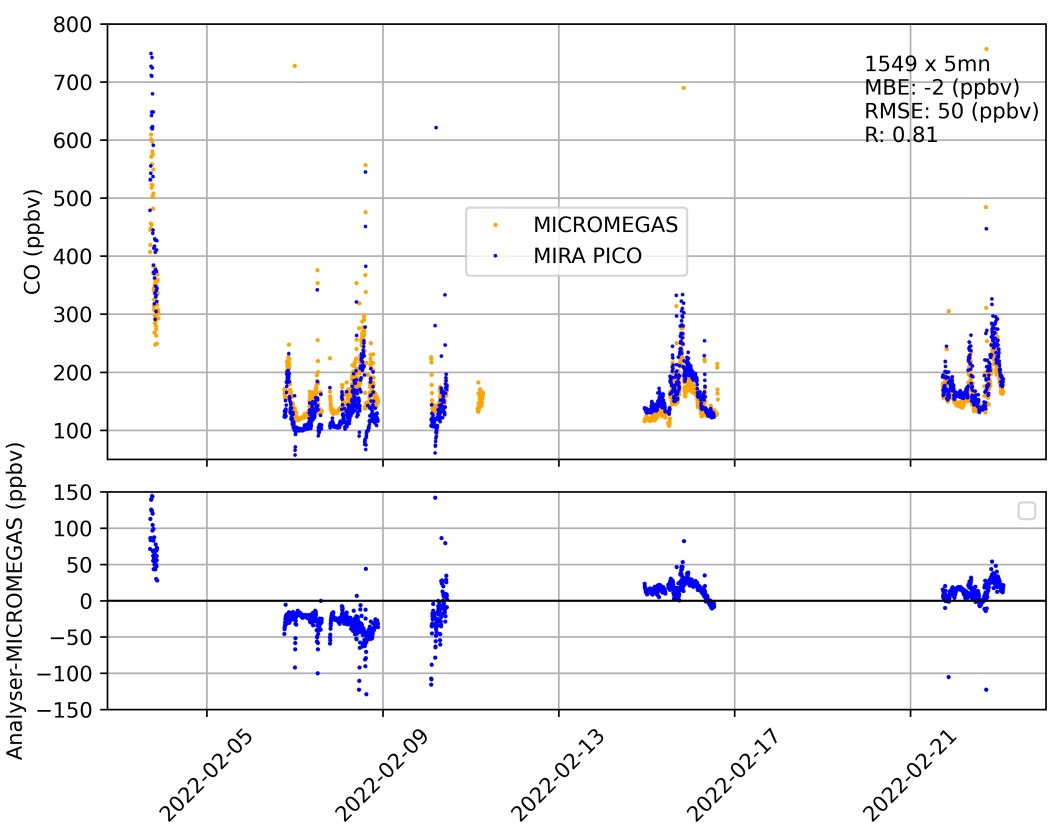

**Figure 12.** Time serie of CO measurements at the UAF-Farm site: (blue symbols) MICROMEGAS data (orange symbols) MoMuCAMS analyser (MIRA PICO) data. The differences <Analyser-MICROMEGAS> are displayed in the bottom panel.



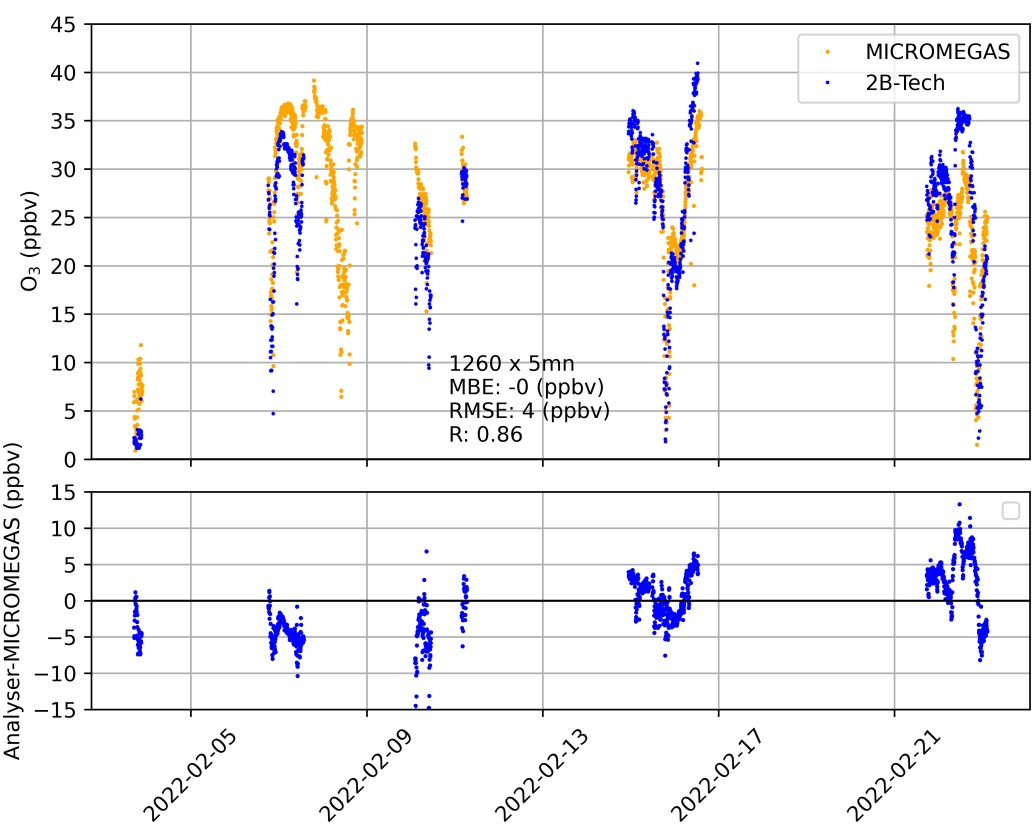

**Figure 13.** Time serie of O$_3$ measurements at the UAF-Farm site: (blue symbols) MICROMEGAS data (orange symbols) MoMuCAMS analyser (2B-Tech) data. The differences <Analyser-MICROMEGAS> are displayed in the bottom panel.



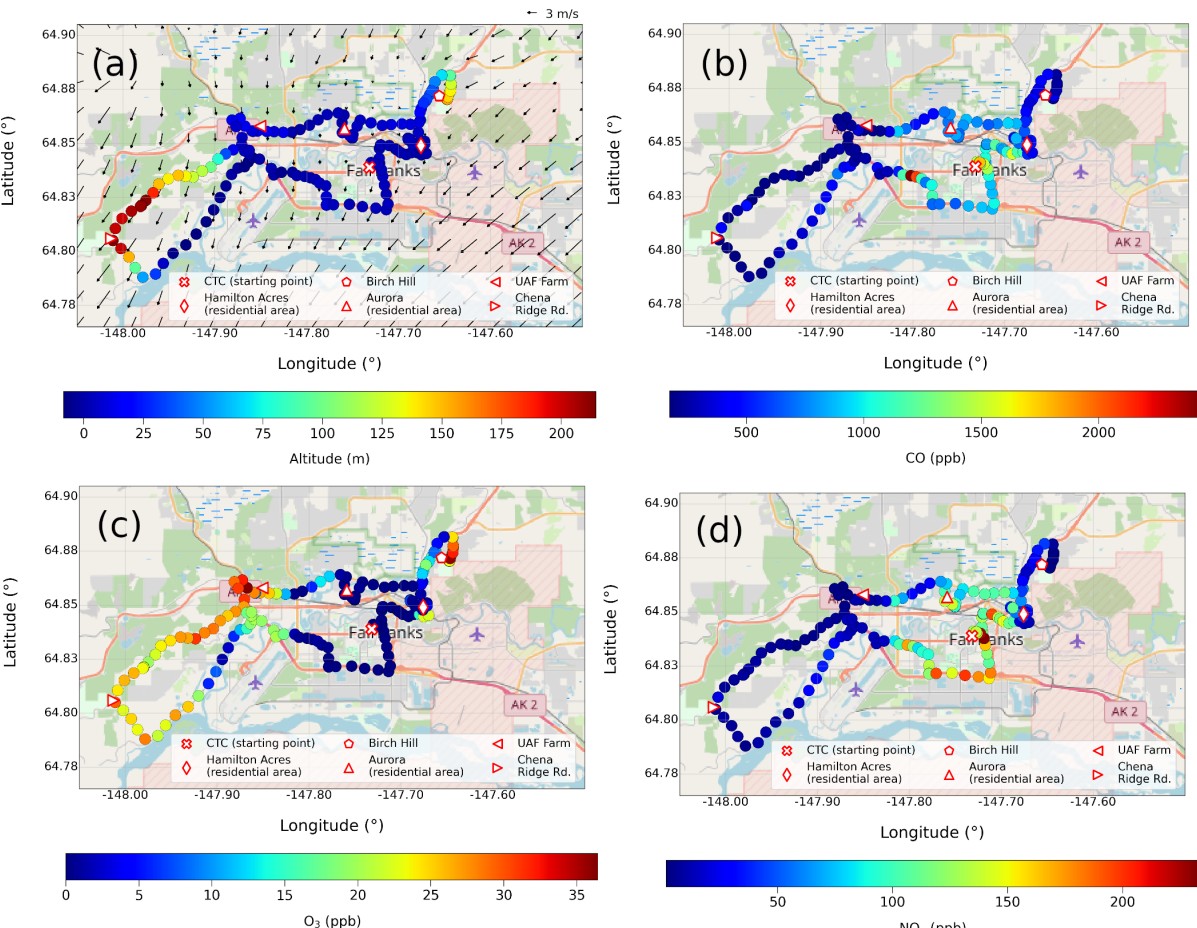

**Figure 14.** Maps with MICROMEGAS measurements from the drive of January 21 for (a) altitude with superimposed 0-10 m winds from the WRF simulation (12-4PM average) (b) CO (c) $O_3$ and (d) NOx. ©OpenStreetMap contributors 2024. Distributed under the Open Data Commons Open Database License (ODbL) v1.0.







**Figure 15.** Time series of (top panel) longitude and altitude (middle panel) NO and $NO_2$ (bottom panel) CO and $O_3$ for the drive of January 21 (minute averages). For each trace gaz, the solid line corresponds to MICROMEGAS measurements and full circles correspond to measurements from the reference analysers at CTC for the two periods at the beginning and at the end of the drive when the car was parked in a street next to the CTC site. The full circles in darker colors correspond to the median MICROMEGAS values during the car stops. The horizontal dashed bars represent the duration of the stop and the vertical solid one the $10^{th}$-$90^{th}$ percentile range. The numbers on top of the NO line in the middle panel identify the stops as follows: (0) CTC (1) Hamilton Acres (2) Birch Hill (3) Aurora (4) UAF-Farm (5) Chena Ridge (6) CTC.





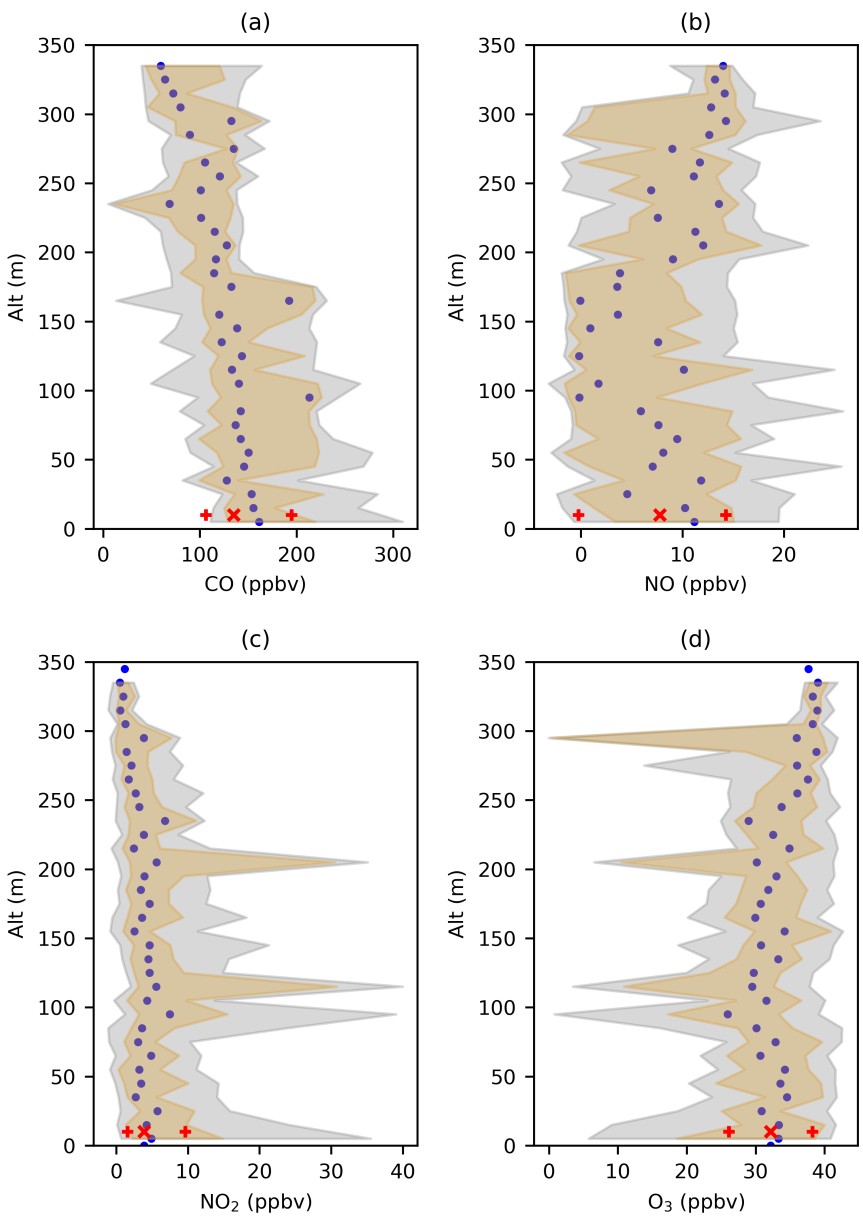

**Figure 16.** Statistics of the vertical profiles of CO, NO, NO$_2$ and O$_3$: (blue dots) median values, orange shaded area: 20-80th percentile range; gray shaded area: 5-95th percentile range. (red x) median value ; (red +) 20 and 80$^{th}$ percentile for the whole Helikite profile dataset.





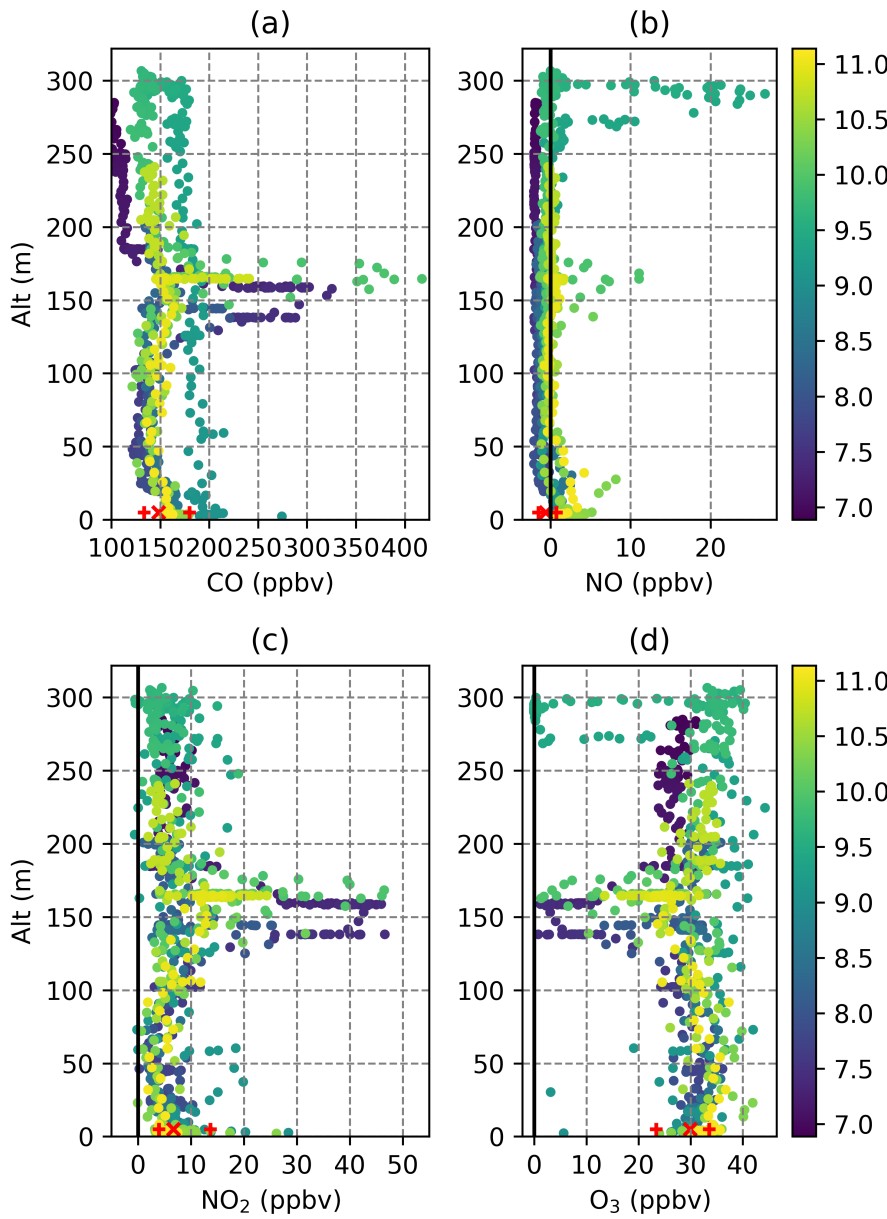

**Figure 17.** MICROMEGAS profiles of (a) CO, (b) NO, (c) NO$_2$ and (d) O$_3$ for the Helikite flight on the morning of February 20. The colors correspond to the hour (Alaskan Time) of observation since the start of the data acquisition. (red x) median value ; (red +) 20 and $80^{th}$ percentile for the whole flight.