# Peer review of "Surface distributions and vertical profiles and of trace gases (CO, O3, NO, NO2) in the Arctic wintertime boundary layer using low-cost sensors during ALPACA-2022"

_EGUsphere, 2024_

## Author Comment (AC1)

**Reply to Review #1:**

*We thank reviewer #1 for acknowledging our work and achievements and for his comments that allowed improving our manuscript. Below are our replies to those comments:*

Line 10, "The correlation coefficient R": only a question at this stage of the review, is it really the R or the R2?

*We provide the correlation or Pearson coefficients R. We have added "Pearson" at the first instances to make sure the reader knows which coefficient we are using.*

Line 18: What does a.g.l. mean?

*We have corrected at the first appearance in the abstract and in the manuscript "above ground level (a.g.l.)".*

Line 20: do you have some data to illustrate the "almost constant"?

*We do not provide details in the abstract but the data corresponding to this statement are discussed in section 3.4 and displayed in Figure 16 with the vertical profiles of the 4 trace gases.*

Line 57: What is the range of temperature?

*The outside temperature is roughly ranging from -35° to 5°C (see Figure 1) but the EGS are in a thermoregulated box with temperatures between 11 and 28°C (see section 2.4 and Figure 3).*

"Figure 3" (Barret et al., 2024, p. 8) Figure 3: If the period without data is not usefull, I would advice the author to use some cut in the time series, or multiple graphs in order to maximise the visibility instead of leaving a third of the space empty.

*It is true that there are some gaps without data but the figure is readable and consistent with the figures 5, 7, 9 and 11. Showing the whole period also allows to better locate the time with measurements at CTC and is also more consistent with Figure 2 displaying the MICROMEGAS operations.*

Line 227, "the addition of voltages from the NO sensor in equ. 2.5.2": What is equ.2.5.2, do you mean equation (1) at the top of the page ?

*The error is corrected.*

Line 237, "correlation coefficient R.": In relation to the comment in the abstract, it is maybe a good idea to write down the formula.

*See reply to comment Line 10. Adding "Pearson" is similar to providing the equation.*

Figure 5: it is not easy to see the difference in color of the dot in the legend (printed or pdf), I would advice to increse the size of the symbol.

*The caption text gives the details about the correspondence between the colors/symbols and the parameters in case the reader has problems with the small symbols of the legend.*

Figure 6(b): could you give some info about the 2 line (I guess red = linear regression, blu = unity)

*Done for Figure, 4, 6, 8 and 10.*

Figure 7: I think some words are missing, "Same caption as Figure 5 but for NO2 (MLP100 calibration function)."

*Corrected for Figures 7,9 and 11.*

Figure 9: Same comment as for Figure 7.

*See above.*

---

## Author Comment (AC2)

**Reply to Review #2:**

*We thank reviewer #2 for her/his in-depth reading  of our manuscript and her/his thorough comments that allowed improving our manuscript. Below are our replies to those comments:*

General comments:

1. Based on the manuscript flow, consider modifying the title to: "Surface Distributions and Vertical Profiles of Trace Gases (CO, $O_3$, NO, $NO_2$) in the Arctic Wintertime Boundary Layer Using Low-Cost Sensors During ALPACA-2022".

*Agreed and corrected.*

2. The study emphasizes the novelty of deploying EGS in cold environments, but the impacts of temperature and relative humidity on sensor performance are not discussed. Please address this in the manuscript.

*The impact of temperature is mitigated using a temperature regulation that maintained the sensor temperature between 11 and 28°C as described in section 2.4. Furthermore the impacts of temperature and humidity are accounted for through the calibration with these parameters part of the learning data. There are no correlation between sensor temperature and air relative humidity variations (Figure 3) and the discrepancies between MICROMEGAS and reference analysers for the validation dataset (Figure 5, 7, 9, 11).*

3. Line 95 mentions $SO_2$ measurement, but no data is presented. Either provide detailed discussion of $SO_2$ data or remove its mention.

*We removed this mention to SO2.*

4. Cross-interference: In lines 225–230, what contributions do NO and $NO_2$ concentrations have on $O_3$ sensor calibration under cold conditions? Would these impacts vary with temperature changes?

*As mentioned above, the sensors are in a temperature regulated box. Therefore, we cannot detect any specific impact of NO and NO2 on O3 measurements specific for cold conditions.*

5. Lines 145–150: Discuss whether vehicle emissions would influence sensor detection. Was the detection of the sensors based on gas diffusion, or was there an air pump? If a pump was used, what was its flow rate? If based on diffusion, would sensor response vary at different vehicle speeds? For example, if the vehicle speed is too fast, the sensor response may not be fast enough to capture the rapid change of ambient concentration. Please discuss.

*As described in the manuscript, the instrument and its sensors are set up in a closed insulated box itself in a bigger plastic container with an inlet pumping outside air with a mini-pump with a constant 0.3 l/mn flow. The vehicle speed variations do therefore not affect the sensors. It has been chosen around 20 m.p.h. in order to provide a horizontal resolution of 500 m for a 1 minute average as stated in the manuscript (l148-149).*

6. Lines 152–155: What were the vertical profiles of temperature, RH, and pressure during balloon ascents? How did rapid changes in these parameters affect sensor performance?

*The sensor temperature variations were limited because of the thermal regulation of the instrument. The slow ascent/descent (20 m/mn or 0.33m/s) and limited altitude reached by the balloon (<350m) prevented fast variations of RH and pressure that could affect the sensor signals. Furthermore, the values are averaged over 15s representing a maximum of 5 m ascent/descent and only 65 Pa pressure variations or 0.065% of the surface pressure.*

7. Lines 152-155: In the balloon measurements, have the authors compared EGS data with high-accuracy instruments (e.g., for CO and $O_3$) to validate the sensor's vertical profiling capabilities?

*We indeed have a limited number of days where the balloon was flown part of the time with MICROMEGAS and either the PICO CO analyzer or the B2-Tech O3 analyzer or both. This was not presented to make the paper lighter. But the reviewer is right, those data provides information which is complementary to the ground comparisons between MICROMEGAS and the analysers on the flight location. We therefore provide a short sub-section (3.2.2) with these comparisons with a table summarizing the statistics (Table 2) and a figure (Figure 14) with comparisons from the only flight with MICROMEGAS and both analysers. The flight dataset is very limited and the time average a lot reduced (15s instead of 300 s at the ground) but the agreement remains very good and even better for CO. For O3 the correlation coefficient is decreased (0.73 compared to 0.86) and the RMSE increased (8 ppbv instead of 4) probably because 15 s averages are much noisier than 300 s one. The data corresponding to the flight of February 23 clearly show that both instruments capture the same vertical variability with enhanced CO and low O3 close to the ground and background conditions aloft.*

8. Figure 3: Why not randomly separate the training and testing datasets? This approach might be more robust and objective.

*In most of the studies, the training and testing datasets are indeed separated randomly and are therefore almost identical. We did otherwise because the instrument was not operated continuously at the calibration site. It was stopped and unplugged, then it performed flights under the balloon or sniffer rounds on the roof of a vehicle around Fairbanks and was then unplugged again, installed on the calibration site for new measurements etc. We therefore preferred to identify different consecutive periods for training and for tests. This way, the test data are more similar to data recorded during flights or sniffer rounds.*
*Nevertheless our first experiment was done with a random selection of learning and prediction data. In that case, all the calibration models are performing very well and very similarly with R generally exceeding 0.95 and very low RMSE for both learning and prediction. This data selection method is therefore not effective for discriminating the best model. As mentioned in the manuscript, using independent training and test datasets allow to identify the models able to extrapolate data outside of the training dataset. The non parametric methods (RF and HGBT) are therefore excluded because they perform worse when applied to the prediction data which is slightly different from the*

*training dataset contrarily to the parametric methods such as the MLP ANN. Our method of selection therefore allows to better select the model best suited for our application. We have added the following text in section 2.5.1 describing the calibration methods to deal with this issue:*

*"To select the most robust method, the calibration data had to be split into two parts, one for the learning of the calibration functions (Equation 1 below), and the second one to validate the predictions made by these functions. The use of randomly selected training and prediction data yields excellent results, with correlation coefficients generally exceeding 0.95 for both training and prediction data. The fact that the performances of the different models are almost identicals, as well as the results obtained from the training and prediction data, does not allow for selecting the best model with this random choice of prediction data. Furthermore, when the trained models are applied to balloon data, some of them produce inconsistent results such as constant gas concentrations during portions of the flights and abrupt transitions from one constant value to another. This is particularly true for the non-parametric methods, RF and HGBT, which struggle to extrapolate beyond the training dataset. Indeed, during balloon flights, weather conditions and especially pollution levels differ from those encountered in the city at the CTC site. In order to overcome those difficulties, the CTC data were split into two equal and independent parts of ~125 hours (see Figure 3) both containing very cold and highly polluted periods and warmer and less polluted periods. The choice of two independent subsets instead of a validation set randomly selected from the whole dataset allows to test methods more robustly, particularly their ability to extrapolate when conditions fall outside the training set limits as will be presented in section 3.1."*

9. Lines 202–205: Add a brief description of the HGBT and RF parameterization methods used.

*We have provided the following details to describe these methods in section 2.5.1:*

*"The RF model is set with 100 estimators (trees in the forest), 5 samples required to be at a leaf node and a minimum of 2 samples required to split an internal node. The HGBT model has a maximum number of iterations of the boosting process, (maximum number of trees) set to 100 and a maximum number of leaves for each tree set to 15."*

10. Line 259: Include a brief explanation of how to interpret the Taylor diagram.

*The statistics used to evaluate the data are detailed in section 5.3.1: Evaluation statistics. Line 239-246 are especially dedicated to explain how to interpret the Taylor diagram. We do not think a new explanation is necessary.*

11. Lines 259–271: Until the end of the manuscript, I realized that Figure 4b presents the information of the raw data, and the raw data for CO was used in this study without any correction. This information should be clearly stated here.

*The raw model was indeed not introduced in the manuscript and we have provided the following description in section 2.5.1:*

*"The calibration model from the manufacturer, called raw in the manuscript, is based on a linear regression (LR) between calibrated gaz concentrations and the voltages output by the electrodes of the sensors."*

12. Lines 292–296: Could the relatively worse performance of HGBT and RF models result from the data selection process? Consider discussing whether a 10-fold cross-validation or random data selection could improve predictions.

*See reply to comment 8 where this issue is dealt with.*

13. Line 296: What is meant by the term "raw calibration method"? Does it refer to using raw data directly or applying a linear correction to the raw data?

*See reply to comment 11.*

14. Line 305: The manuscript lacks information on Ox calibration. Please include this.

*With this comment we understand that there was some confusion in the manuscript with some statements based on older versions of the calibration method used for O3 that were not removed. O3 is calibrated directly with the data from the O3 analyser with the same equation than the other gases with voltages from the Ox, NO2 and NO sensors. We therefore do not calibrate Ox nor apply O3 = Ox – NO2.*
*The manuscript has been changed accordingly with lines 228-229 changed from :*
*"For Ox the best results are obtained with the addition of the voltages of the NO2 and also of the NO sensor. The concentration of O3 is then obtained by substracting [NO 2 ] from [O x ]." to*
*"For O3 the best results are obtained with the addition of the voltages of the NO2 and also of the NO sensor."*
*And statement "Once Ox and NO2 are calibrated, we compute O3 = Ox - NO2" has been removed.*

15. Line 327: Define what is meant by "systematic bias."

*We meant "significant bias" which means that the bias is larger than the 1sigma or standard deviation. We have changed the text.*

16. Figure 15: Was the time resolution of sensor data consistent with that of the reference instrument? If not, could observed spikes in sensor data be due to its higher time resolution?

*Both data are averaged over 1 minutes as for the calibration. The sensor data are recorded at 1Hz and at this frequency the data are much noisier but do not present spikes. The peaks of CO observed with the sensor are not instrumental spikes but come from the observation onboard a vehicle parked on the road compared to an instrument with its inlet on top of a trailer some 10 meters away from the road as explained in the manuscript.*

17. Section 4: Discuss the limitations of this study and please provide suggestions for future research on low-cost sensors in cold environments.

*Some limitations and suggestions were already discussed in the conclusion but we added two important suggestions for future use of LC sensors in extremely cold conditions:*

*"The sensors were thermoregulated in an insulated box with a simple and lightweight system based on a thermal switch and a thin heating film. Nevertheless, they are supposed to operate down to very low temperatures. It would be interesting to use them without thermal regulation to determine whether they can still function properly with the calibration method presented in this study. This could allow to further reduce the system's weight, size and energy consumption."*

*"[…] calibration with high-quality reference data is the crucial step for obtaining accurate measurements. This is probably the most significant limitation in using these sensors. Our study demonstrated that meaningful results could be achieved with the appropriate calibration method, even beyond the calibration dataset limits. It would be interesting to observe the sensors' behavior in areas further away from urban sources, where concentration variations are low around background levels."*

Technical comments:

1. Line 57: Extremely, not extreme. => *corrected*
2. Line 142: Correct "ouside" to "outside." => *corrected*
3. Line 208: Figure 7 is referenced before Figure 6. Correct the order. => *Figure 7 corresponds to the section dedicated to NO calibration and the order cannot be changed. And it is the best figure to show the pollution levels for learnig and prediction periods. So we kept it like it is even though it is not orthodox.*
4. Line 227: Replace "Equ." with "Eqn. 2.5.2." => *changed with Equation 1*
5. Figure 15: Trace gas, not trace gaz, in the caption. For better visualization, consider using a different symbol instead of dark circles. => *changed to Trace. We find the full color circles have a rather good visibility.*